# Revisiting Score Propagation in Graph Out-of-Distribution Detection

**Longfei Ma**[1*], **Yiyou Sun**[2*], **Kaize Ding**[3], **Zemin Liu**[1†], **Fei Wu**[1†]
[1]Zhejiang University, [2]University of Wisconsin-Madison, [3]Northwestern University
{longfeima, liu.zemin, wufei}@zju.edu.cn,
sunyiyou@cs.wisc.edu, kaize.ding@northwestern.edu

## Abstract

The field of graph learning has been substantially advanced by the development of deep learning models, in particular graph neural networks. However, one salient yet largely under-explored challenge is detecting Out-of-Distribution (OOD) nodes on graphs. Prevailing OOD detection techniques developed in other domains like computer vision, do not cater to the interconnected nature of graphs. This work aims to fill this gap by exploring the potential of a simple yet effective method – OOD score propagation, which propagates OOD scores among neighboring nodes along the graph structure. This post hoc solution can be easily integrated with existing OOD scoring functions, showcasing its excellent flexibility and effectiveness in most scenarios. However, the conditions under which score propagation proves beneficial remain not fully elucidated. Our study meticulously derives these conditions and, inspired by this discovery, introduces an innovative edge augmentation strategy with theoretical guarantee. Empirical evaluations affirm the superiority of our proposed method, outperforming strong OOD detection baselines in various scenarios and settings. To ensure reproducibility, we have made our code and relevant data publicly available at https://github.com/longfei-ma/GRASP.

## 1 Introduction

Graph-like data structures are ubiquitous in many domains, such as social networks [87, 40], molecular chemistry [19, 82], and recommendation systems [84, 45]. As graph neural networks increasingly serve as powerful tools for navigating this complex data landscape, a compelling yet under-explored issue emerges: Out-of-Distribution (OOD) node detection. Imagine a recommender system suggesting irrelevant or even harmful products to users, or a bioinformatics algorithm misusing an unknown protein. This gives rise to the importance of OOD detection in graph data, which determines whether an input is in-distribution (ID) or OOD and enables the model to take precautions.

While existing OOD detection methods have shown promising results in computer vision [66, 27, 18, 15, 95], natural language procession [11, 57] and tabular data analytics [68], their effectiveness diminishes when applied to graph data [77]. These conventional techniques operate under the assumption that data points are independently sampled, which misaligns with the interconnected nature of graphs. To better leverage the structural knowledge from the graph, OOD score propagation [77] has been employed to enhance graph OOD detection performance by directly propagating the computed OOD scores along the graph structure (as shown in Figure 1). Although this strategy has shown promising results on some datasets,

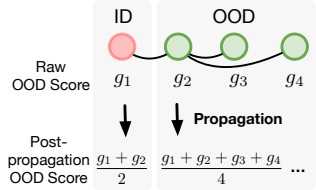

Figure 1: Illustration of OOD scores propagation.

---

*Equal Contribution

†Corresponding author

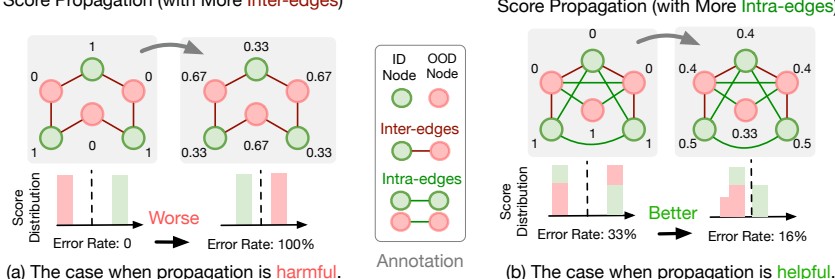

Figure 2: Two illustrative examples when scoring propagation is harmful/helpful. We consider ID nodes in green and OOD nodes in red. Inter-edges are defined to be `ID-to-OOD` edges and Intra-edges are `ID-to-ID` or `OOD-to-OOD` edges. The value represents the respective OOD scores. Consequently, the propagated scores in these cases will be the mean of the scores of adjacent nodes as shown in Figure 1.

the reasons behind its effectiveness and the conditions under which it works are not clear. To dive deep into it, our research embarks on addressing two research questions related to OOD score propagation:

*Question 1: "Will naive OOD score propagation always help graph OOD Detection?"* The short answer is no. This method can be ineffective on graphs where inter-edges (`ID-to-OOD`) are predominant. Using the examplse in Figure 2(a), where only inter-edges exist, prior to conducting score propagation, all ID and OOD nodes can be fully distinguished. However, after performing score propagation along these edges, the ID and OOD nodes are completely misclassified. Conversely, after adding intra-edges and making them donimate, as shown in Figure 2(b), score propagation would be beneficial to distinguish ID and OOD nodes. These two examples intuitively illustrate how the ratio of intra-edges and inter-edges can impact the effectiveness of OOD score propagation. We substantiate this intuition in Section 3. This finding naturally paves the way for subsequent questions.

*Question 2: "How to derive a better score propagation strategy for graph OOD detection?"* Building on our prior findings, we propose a graph augmentation strategy as presented in Section 4. Specifically, our strategy selects a subset $G$ of the training set and puts additional edges to the nodes within $G$. Beyond its practical implications, our solution is theoretically supported: When $G$ predominantly connects to ID data over OOD data, our strategy can provably enhance the post-propagation OOD detection outcomes.

We summarize our contributions as below:

- **Theoretical understanding:** We delve deeply into the mechanism of score propagation to understand its potential for graph OOD detection and elucidate the conditions under which it thrives, providing an understanding that extends beyond existing knowledge.
- **Practical solution:** To counter the identified challenge of inter-edges' domination, we propose **GR**aph-**A**ugmented **S**core **P**ropagation (GRASP), an innovative edge augmentation strategy with theoretical guarantee. By strategically adding edges to a chosen subset $G$ of the training set, as detailed in Section 4, our method aims to enhance the intra-edge ratio, thereby boosting OOD detection outcomes post-propagation.
- **Empirical studies**: We demonstrate the superior performance of the proposed method on extensive graph OOD detection benchmarks, different pre-trained methodologies [34, 69, 7, 94], and different OOD scoring functions. Under the same condition, our proposed strategy substantially reduces the FPR95 by **17.87**% and **32.21**% compared to the strongest graph OOD detection baselines on common and large-scale benchmarks respectively. Comprehensive analyses are also provided to validate the effectiveness of the proposed approach and the correctness of the theoretical findings.

## 2 Preliminaries

**Problem setup**. We consider a traditional semi-supervised node classification setting with the additional unlabeled nodes from the out-of-distribution class. Let $\mathcal{G} = \{\mathcal{V}, \mathcal{E}\}$ denote the graph with nodes $\mathcal{V}$ and edges $\mathcal{E}$, where the node set $\mathcal{V}$ with size $N$ are attributed with data matrix $X \in \mathbb{R}^{N \times d}$. The structure of graph $\mathcal{G}$ is described by the adjacency matrix $A \in \{0, 1\}^{N \times N}$. We let the corresponding row-stochastic matrices as $\bar{A} = \mathrm{D}^{-1}A$, where $D$ is the diagonal matrix with

$D_{ii} = \sum_j A_{ij}$. The $N$ nodes are partially labeled, so we let $\mathcal{V}_l$ and $\mathcal{V}_u$ represent the labeled and unlabeled node sets respectively, i.e, $\mathcal{V} = \mathcal{V}_l \cup \mathcal{V}_u$. Given a training set $\mathcal{D}^{tr} = \left\{ (\mathbf{x}_i, y_i) \right\}_{i \in \mathcal{V}_l}$ with $\mathbf{x}_i$ as the $i$-th row of $X$ and $y_i \in \mathcal{Y} \triangleq \{1, \cdots, C\}$, the goal of node classification is to learn a mapping $f : \mathcal{V} \to \mathbb{R}^C$ from the nodes to the probability of each class.

**Out-of-distribution detection.** When deploying a model in the real world, a reliable classifier should not only accurately classify known in-distribution (ID) nodes, but also identify "unknown" nodes or OOD nodes. Formally, we can represent the unlabeled node set by $\mathcal{V}_u = \mathcal{V}_{uid} \cup \mathcal{V}_{uood}$ where $\mathcal{V}_{uid}$ and $\mathcal{V}_{uood}$ represent the in-distribution (ID) node and OOD node respectively. **The goal of the graph OOD detection** is to derive an algorithm to decide if a node $i \in \mathcal{V}_u$ is from $\mathcal{V}_{uood}$ or $\mathcal{V}_{uid}$.

This can be achieved by having an OOD detector, in tandem with the node classification model $f$. OOD detection can be formulated as a binary classification problem. At test time, the goal of OOD detection is to decide whether an unlabeled node $i \in \mathcal{V}_u$ is from ID or OOD. The decision can be made via a level set estimation:

$$F_{OODD}(i, \mathcal{G}; \lambda) = \begin{cases} \text{ID} & g(\mathbf{x}_i) \geq \lambda \\ \text{OOD} & g(\mathbf{x}_i) < \lambda \end{cases},$$

where nodes with higher scores $g(\mathbf{x}_i)$ are classified as ID and vice versa, and $\lambda$ is the threshold commonly chosen so that a high fraction (e.g., 95%) of ID data is correctly classified.

In this paper, we consider **post hoc** OOD detection methods to produce $g(\mathbf{x}_i)$ which does not require expensive re-training. As an example, a classical way to compute $g(\mathbf{x}_i)$ is Maximum Softmax Probability (MSP) [23] which is given by the maximum softmax value. We include details of the considered OOD detection methods in Appendix B.

## 3   Will propagation always help Graph OOD Detection?

In the introduction, we delineate the limitations of OOD score propagation using a concrete example and elucidate the intuition that it may fail when inter-edges dominate. In this section, we formally delineate the conditions under which OOD score propagation works. We start by showing the formal definition of propagation.

**Define OOD scoring propagation**. Given a raw OOD scoring vector $\hat{\mathbf{g}} \in \mathbb{R}^N$ with $\hat{\mathbf{g}}_i = g(\mathbf{x}_i)$, the propagated scoring vector is given by:

$$\text{Propagated OOD Scoring Vector: } \mathbf{g} = \bar{A}^k \hat{\mathbf{g}}, \tag{1}$$

where $k \in \mathbb{N}^+$ are hyperparameters.

*Is it necessarily the case that* $\mathbf{g}$ *outperforms* $\hat{\mathbf{g}}$*?* The answer is **NO**. We elucidate with the theoretical insight below.

**Theoretical Insight.** As discussed in the Introduction from Figure 2, when the number of `ID-to-ID` and `OOD-to-OOD` edges surpasses that of `ID-to-OOD` edges, the propagation mechanism tends to "aggregate" the scores associated with the ID and OOD nodes respectively, which further amplify the separability between them. Conversely, when the number of `ID-to-OOD` edges are more than the other types of edges, the scores for both ID and OOD nodes become undistinguishable post-propagation.

The example above offers the insight that the relative performance of $\mathbf{g}$ compared to $\hat{\mathbf{g}}$ is contingent upon the structural dynamics of the network, specifically the distribution of edges. To formally articulate this relationship, we adopt a probabilistic framework for modeling edges. Specifically, we assume that the edge follows a Bernoulli distribution characterized by parameters $\eta_{intra}$ and $\eta_{inter}$ for intra-edges (`ID-to-ID` and `OOD-to-OOD`) and inter-edges (`ID-to-OOD`), respectively:

$$A_{ij} \sim \begin{cases} Ber(\eta_{intra}), & \text{if } i, j \in \mathcal{V}_{uid} \text{ or } i, j \in \mathcal{V}_{uood} \\ Ber(\eta_{inter}), & \text{if } i \in \mathcal{V}_{uid}, j \in \mathcal{V}_{uood} \text{ or } j \in \mathcal{V}_{uid}, i \in \mathcal{V}_{uood} \end{cases}$$

In the context of probabilistic modeling, the subsequent Theorem 3.1 can be established to formalize the inherent understanding.

**Theorem 3.1.** *(Informal) (a) When $\eta_{intra} \gg \eta_{inter}$, it is highly likely that the propagation algorithm will yield enhanced performance in OOD detection. (b) When $\eta_{intra} \approx \eta_{inter}$ or even $\eta_{intra} < \eta_{inter}$, the score propagation is likely to be either ineffective or detrimental to the performance.*

We also provide the formal version below (Theorem 3.2) which provides a mathematical foundation for understanding how varying the Bernoulli parameters influence the efficacy of the propagation in the context of OOD detection. We provide the detailed proof in Appendix A.

**Theorem 3.2.** *(Formal) For any two test ID/OOD node set $S_{id} \subset \mathcal{V}_{uid}, S_{ood} \subset \mathcal{V}_{uood}$ with equal size $N_s$, let the ID-vs-OOD separability $\mathcal{M}_{sep}$ defined on an OOD scoring vector $\hat{\mathbf{g}} \in \mathbb{R}^N$ as*

$$\mathcal{M}_{sep}(\hat{\mathbf{g}}) \triangleq \mathbb{E}_{i \in S_{id}} \hat{\mathbf{g}}_i - \mathbb{E}_{j \in S_{ood}} \hat{\mathbf{g}}_j.$$

*If $\mathcal{M}_{sep}(\hat{\mathbf{g}}) > 0$ and $\eta_{intra} - \eta_{inter} > 1/N_s$, for some $\epsilon > 0$ and constant c, we have $\mathbb{P}\left(\mathcal{M}_{sep}(A\hat{\mathbf{g}}) \geq \mathcal{M}_{sep}(\hat{\mathbf{g}}) - \epsilon\right) \geq 1 - exp(-\frac{c\epsilon^2}{\|\hat{\mathbf{g}}\|_2^2})$.*

**Summary.** This section has presented a comprehensive theoretical evidence to substantiate the claim that propagation through the adjacency matrix $A$ does not necessarily enhance out-of-distribution (OOD) detection in graphs. Moreover, Theorem 3.2 reveals that the critical factor in enhancing post-propagation performance lies in **improving the ratio of intra-edges** within the graph structure. These insights serve as a direct motivation for the augmentation strategy in the next section.

## 4 An Augmented Score Propagation Strategy

The findings from the preceding section give rise to a subsequent thought: "*Can we improve the propagation strategy for graph OOD detection performance?*" In an ideal scenario, if an oracle were to indicate that a particular subset in the test set belongs exclusively to the ID or OOD, one could augment the graph by adding intra-edges or removing inter-edges. This would consequently improve the ratio of intra-edges $\eta_{intra}$, leading to enhanced OOD detection performance post-propagation.

However, such an oracle does not exist in practical settings, and even approximating such a subset proves to be a difficult task. Existing literature has suggested the use of pseudo-labels assigned to nodes [36, 79, 2, 73, 55]. Nonetheless, these studies also caution that this approach is susceptible to "confirmation bias", whereby errors in estimation are inadvertently amplified.

To circumvent it, this paper proposes the solution for adding edges to a subset of the training set $\mathcal{V}_l$, which is assured to be in-distribution data. We start by showing the theoretical underpinnings that adding such a subset can, under specified conditions, contribute to improved OOD detection performance after propagation.

### 4.1 Theoretical Insight

Our approach involves adding the edges to a subset $G$ of training data and then propagating the out-of-distribution (OOD) scoring vector using the enhanced adjacency matrix. Specifically, when edges are added to $G$, this action can be mathematically represented as incorporating a perturbation matrix $E = \mathbf{e}_G \mathbf{e}_G^\top$ into A, as demonstrated in Figure 3. Here, $\mathbf{e}_S \in \mathbb{R}^N$ denotes an indicator vector for a set $S \subset \mathcal{V}$, where the vector takes the value of 1 if the index $i \in S$ and value 0 otherwise. A sufficient condition for the efficacy of this augmentation strategy in enhancing post-propagation OOD detection performance is outlined in Theorem 4.1.

**Theorem 4.1.** *(Informal) For a subset $G$ in the training set, augmenting $G$ by adding edges to all its nodes can lead to improved post-propagation OOD detection performance, provided that the following condition is met: $\underline{G \text{ has more edges to ID data than OOD data.}}$*

We also provide the formal version below (Theorem 4.2) that incorporates a perturbation analysis. This analysis elucidates how edge augmentation in the training set can positively influence the

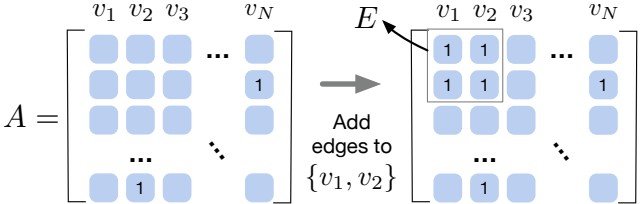

Figure 3: The augmentation procedure.

propagation algorithm's ability to enhance OOD detection. For the sake of the main intuition, we provide the analysis on $A$ instead of $\bar{A}$ for simplicity. We provide the detailed proof in Appendix A.

**Theorem 4.2.** *(Formal) For any two test ID/OOD node set $S_{id} \subset \mathcal{V}_{uid}, S_{ood} \subset \mathcal{V}_{uood}$ with size $N_s$, let the ID-vs-OOD separability $\mathcal{M}_{sep}$ defined on a non-negative OOD scoring vector $\hat{\mathbf{g}} \in \mathbb{R}^N$ as*

$$\mathcal{M}_{sep}(\hat{\mathbf{g}}) \triangleq \mathbb{E}_{i \in S_{id}} \hat{\mathbf{g}}_i - \mathbb{E}_{j \in S_{ood}} \hat{\mathbf{g}}_j.$$

*Let $\mathcal{E}_{S \leftrightarrow S'} \subset \mathcal{E}$ to denote the edge set of edges between two node sets $S$ and $S'$, where $S, S' \subset \mathcal{V}$. If we can find a node set $G \subset \mathcal{V}_l$ such that $|\mathcal{E}_{G \leftrightarrow S_{id}}| > |\mathcal{E}_{G \leftrightarrow S_{ood}}|$, we have*

$$\mathcal{M}_{sep}((A + \delta E)^2 \hat{\mathbf{g}}) > \mathcal{M}_{sep}(A^2 \hat{\mathbf{g}}),$$

*where $E = \mathbf{e}_G \mathbf{e}_G^\top$ and $\delta > 0$.*

The Theorem 4.2 shows a critical principle for enhancing propagation: the optimal strategy entails the addition of edges to the subset $G$ such that there are more edges to ID data than OOD data. For some $S_{id}, S_{ood}$ in the test set, the goal is to find the set

$$G_* = \operatorname*{arg\,max}_{S \subset \mathcal{V}_l, |S| = N_g} \frac{|\mathcal{E}_{S \leftrightarrow S_{id}}|}{|\mathcal{E}_{S \leftrightarrow S_{ood}}|}, \tag{2}$$

where $N_g$ is a hyperparameter to control the size of $G_*$. Inspired by the optimization target, we proceed to present our pragmatic algorithmic approach.

### 4.2 Graph-Augmented Score Propagation (GRASP)

Our augmentation approach hinges on the selection of a subset, $G$, from the training set, as exemplified in Equation 2. Two principal challenges arise in implementing this: (1) We cannot directly determine the number of edges linked to ID/OOD data because these reside in the test set and their labels remain unknown. (2) An exhaustive search to find a subset is computationally expensive, as the number of combinatorial possibilities increases in a factorial manner. In this paper, we tackle these challenges by providing the practical approximation method.

**Selection of $\mathbf{S_{id}}/\mathbf{S_{ood}}$.** Our discussion begins by detailing the methodology to select the subset from the test ID/OOD dataset, symbolized by $S_{id}$ and $S_{ood}$ in Equation 2. A straightforward approach to obtain the most likely ID is by selecting nodes with the largest confidence and the least for OOD in class predictions. Following [23], we employ the max softmax probability (MSP) as a representation of confidence. The selected sets can be defined as:

$$S_{id} = \{i \in \mathcal{V}_u| \max_{c \in [C]} f_c(i) > \lambda_\alpha\}, S_{ood} = \{j \in \mathcal{V}_u| \max_{c \in [C]} f_c(j) < \lambda_{100-\alpha}, \tag{3}$$

where $\lambda_\alpha$ denotes the $\alpha$-th percentile of the MSP scores corresponding to nodes in $\mathcal{V}_u$. To offer a clear view, Figure 4 portrays $S_{id}$ and $S_{ood}$ in the marginal regions highlighted in orange. Selecting a subset in the leftmost and rightmost regions reduces the error when identifying the ID/OOD subsets, given that overlapping between ID and OOD predominantly occurs around the central region of the distribution.

**Selection of $G$.** Upon establishing $S_{id}$ and $S_{ood}$, the next step is to determine $G$ using Equation 2. Directly enumerating every possible $G$ is impractical. Instead, we adopt a greedy approach, prioritizing the node with the highest "likelihood" score. To elucidate, for each node $i \in \mathcal{V}_l$, the score can be

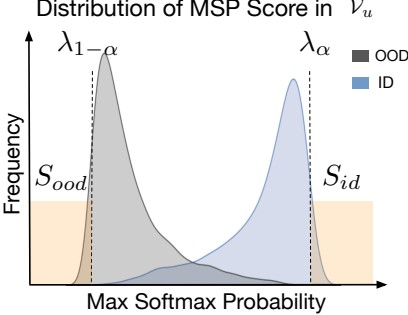

Figure 4: Illustration of the rationale in selecting $S_{id}$ and $S_{ood}$. MSP score is reported on Dataset `Coauther-CS` with the division of ID and OOD classes introduced in Appendix C.

computed as the ratio of the edge count to $S_{id}$ over $S_{ood}$:

$$h(i) = |\mathcal{E}_{\{i\}\leftrightarrow S_{id}}|/(|\mathcal{E}_{\{i\}\leftrightarrow S_{ood}}| + 1), \tag{4}$$

where we incorporate an addition of 1 in the denominator to circumvent division by zero. Subsequently, $G$ can be expressed as:

$$G = \{i \in \mathcal{V}_l | h(i) > \tau_\beta\}, \tag{5}$$

where $\tau_\beta$ stands for the $\beta$-th percentile of $h(i)$ scores for nodes in $\mathcal{V}_l$. Once $G$ is defined, edge augmentation can be executed as demonstrated in Section 4.1. The OOD score is then propagated with the new adjacency matrix $A_+ = A + \mathbf{e}_G \mathbf{e}_G^\top$ in place:

$$\mathbf{g}_{GRASP} = (\bar{A}_+)^k \hat{\mathbf{g}}, \tag{6}$$

where $k \in \mathbb{N}^+$ are hyperparameters.

**Complexity analysis.** While our algorithm introduces the fully connected matrix $E$, our method can be effciently implemented by matrix-vector multiplication, leading to computational footprint in terms of runtime and memory usage with $\mathcal{O}(N + 2k|\mathcal{E}| + n)$ and $\mathcal{O}(N + |\mathcal{E}| + n)$ respectively after propagation for $k$ times, where $n$ is node count of subset $G$. We provide the comprehensive complexity analysis in Appendix D.6.

## 5 Experiments

**Datasets.** We conduct extensive experiments using 10 real-world datasets that span diverse domains, scales, and structures (homophily or heterophily). A high-level summary of the dataset statistics is provided in Table 1, with a detailed information of the datasets and the comprehensive description of ID/OOD split in Appendix C. Specifically, `Cora` [61] serves as a widely recognized citation network. `Amazon-Photo` [52] represents a co-purchasing network on Amazon. `Coauthor-CS` [62] portrays a coauthor network within the realm of computer science. Moreover, `Chameleon` and `Squirrel` [59] are two notable Wikipedia networks, predominantly utilized as heterophilic graph benchmarks. We

Table 1: Summary statistics of the datasets: size of the training set $|\mathcal{V}_l|$, test ID set $|\mathcal{V}_{uid}|$, test OOD set $|\mathcal{V}_{uood}|$; number of ID classes $C$, scale of the dataset, and whether the graph is homophily.

| Dataset | $|\mathcal{V}_l|$ | $|\mathcal{V}_{uid}|$ | $|\mathcal{V}_{uood}|$ | $C$ | Scale | Homph |
|---|---|---|---|---|---|---|
| Cora | 180 | 724 | 18K | 3 | SM | ✓ |
| Amazon-Photo | 618 | 2K | 4K | 3 | SM | ✓ |
| Coauthor-CS | 2K | 10K | 5K | 11 | SM | ✓ |
| Chameleon | 272 | 1K | 916 | 3 | SM | ✗ |
| Squirrel | 622 | 2K | 2K | 3 | SM | ✗ |
| Reddit2 | 33K | 133K | 65K | 11 | LG | ✓ |
| ogbn-products | 130K | 522K | 1M | 11 | LG | ✓ |
| ArXiv-year | 23K | 92K | 53K | 3 | LG | ✗ |
| Snap-patents | 351K | 1M | 1M | 3 | LG | ✗ |
| Wiki | 212K | 850K | 862K | 3 | LG | ✗ |

additionally incorporate 5 **large-scale** graphs to evaluate our proposed methods: `Reddit2` [88] and `ogbn-products` [24] are large homophily datasets; `ArXiv-year`, `Snap-patents`, and `Wiki` [44] are recently proposed large-scale heterophily benchmarks.

**Remark on homophily/heterophily.** In Table 1, datasets are also categorized based on the attribute of homophily, denoting the tendency of nodes with the same class to connect. Conversely, the heterophily graph demonstrates a tendency for nodes of disparate classes to connect. This characteristic not only presents a challenge for node classification but also for graph OOD detection. The underlying

Table 2: **Main results on common benchmarks.** Comparison with competitive out-of-distribution detection methods on pre-trained GCN. We take the average values that are percentages over 5 independently trained backbones. ↑ (↓) indicates larger (smaller) values are better.

| Method | Cora | | Amazon | | Datasets Coauthor | | Chameleon | | Squirrel | | Average | |
|---|---|---|---|---|---|---|---|---|---|---|---|---|
| | FPR↓ | AUROC↑ | FPR↓ | AUROC↑ | FPR↓ | AUROC↑ | FPR↓ | AUROC↑ | FPR↓ | AUROC↑ | FPR↓ | AUROC↑ |
| MSP | 70.86 | 84.56 | 49.26 | 89.34 | 28.82 | 94.34 | 85.70 | 57.96 | 94.68 | 48.51 | 65.86 | 74.94 |
| Energy | 67.54 | 85.47 | 42.13 | 90.28 | 20.29 | 95.67 | 88.06 | 59.20 | 93.98 | 45.07 | 62.40 | 75.14 |
| KNN | 90.20 | 70.94 | 65.19 | 84.71 | 51.24 | 90.13 | 93.38 | 57.90 | 94.72 | 54.68 | 78.95 | 71.67 |
| ODIN | 68.41 | 84.98 | 44.06 | 89.90 | 22.59 | 95.27 | 85.31 | 57.94 | 94.17 | 44.08 | 62.91 | 74.43 |
| Mahalanobis | 69.68 | 85.48 | 96.49 | 75.58 | 85.71 | 84.98 | 95.55 | 53.19 | 94.90 | 54.99 | 88.47 | 70.84 |
| GKDE | 63.71 | 86.27 | 81.29 | 77.26 | 25.48 | 95.13 | 92.93 | 50.14 | 96.71 | 49.38 | 72.02 | 71.64 |
| GPN | 58.45 | 82.93 | 72.95 | 82.63 | 34.11 | 93.82 | 82.25 | 68.20 | 95.58 | 48.38 | 68.67 | 75.19 |
| OODGAT | 94.59 | 53.63 | 71.34 | 66.95 | 96.53 | 52.18 | 94.43 | 59.67 | 95.27 | 46.13 | 90.43 | 55.71 |
| GNNSafe | 54.71 | 87.52 | 22.39 | 96.27 | 16.64 | 95.82 | 100.00 | 50.42 | 100.00 | 35.88 | 58.75 | 73.18 |
| **GRASP (Ours)** | **29.70** | **93.50** | **14.38** | **96.68** | **7.84** | **97.75** | **66.88** | **76.93** | **85.59** | **61.09** | **40.88** | **85.19** |

reason is that the OOD data is from different classes with ID, and heterophily exacerbates the ratio of inter-edge connections between ID and OOD, which is deemed undesirable for graph OOD detection according to Theorem 3.2.

**Implementation Details.** Our graph OOD detection technique operates in a *post hoc* fashion utilizing a pre-trained network and so can be used in various pre-trained network seamlessly. We present results evaluated on Graph Convolutional Network (GCN) [34] in the main paper to save space and put detailed results of other architectures in Appendix D.3. All pre-trained models possess a layer depth of 2. With the pre-trained network, we proceed to execute the graph OOD detection. By default, we report the performance of the augmented propagation (GRASP) on the MSP score [23]. The compatibility with other OOD scoring functions is also shown in Table 6. We set the propagation number $k$ as 8, with percentile values $\alpha = 5$ and $\beta = 50$. The sensitivity analysis of the hyper-parameters is included in Appendix D.4.

**Metrics.** Following the convention in literature [23, 47, 65], we use AUROC and FPR95 as evaluation metrics for OOD detection.

## 5.1 Main Results

**GRASP consistently achieves superior performance.** We provide results of 5 common small-scale benchmarks and 5 **large-scale** datasets in Table 2 and Table 3 respectively, wherein only the averaged results over 5 runs are presented to save space and the detailed results with standard errors of these two scale datasets are shown in Table 12 and Table 13 respectively. From the results we can see that our proposed methodology (GRASP) consistently demonstrates promising performance. The comparative analysis encompasses a broad spectrum of *post hoc* competitive Out-of-Distribution (OOD) detection techniques in existing literature and training-based methods tailored for graph OOD detection. We categorize the baseline methods into two groups: (a) Traditional OOD detection methods including MSP [23], Energy [47], ODIN [43], Mahalanobis [37], and KNN [67]; (b) Graph OOD detection methods including GKDE [92], GPN [64], OODGAT [63], and GNNSafe [77]. In these tables, we present GRASP results based on the MSP score. Noteworthy findings include: (a) The traditional OOD detection methods exhibit suboptimal performance in the realm of graph OOD detection. For instance, GRASP reduced the average FPR95 by **17.87**% and **32.21**% compared to the strongest traditional OOD detection method GNNSafe and Mahalanobis on common and large-scale benchmarks, respectively. This outcome is anticipated given their lack of specificity in design towards graph data. (b) GRASP outperforms existing baselines by a large margin, surpassing the best baseline GNNSafe by **17.87**% and **40**% concerning average FPR95 on two scale benchmarks respectively. These results further corroborate that the theoretically motivated solution GRASP is also appealing to use in practice.

**GRASP is also competitive on large-scale graph datasets.** Contrasted with the small-scale benchmarks in Table 2, the large-scale scenario in Table 3 presents more challenges due to a large number of nodes and edges. From Table 3 we can see that all baseline OOD detection methodologies exhibit suboptimal performance on large-scale benchmarks, while our method GRASP robustly performs the best.

**GRASP exhibits significant advantages over training-based baselines.** In addition to contrasting with *post hoc* methods, we extend our comparison to a parallel line of graph Out-Of-Distribution

Table 3: **Main results on large-scale benchmarks.** Comparison with competitive out-of-distribution detection methods on pre-trained method GCN. We take the average values that are percentages over 5 independently trained backbones. OOM means Out-Of-Memory and OOT denotes that no results have been got after running over 48 hours for each run. ↑ (↓) indicates larger (smaller) values are better.

| Method | reddit2 FPR↓ | reddit2 AUROC↑ | ogbn-products FPR↓ | ogbn-products AUROC↑ | arxiv-year FPR↓ | arxiv-year AUROC↑ | snap-patents FPR↓ | snap-patents AUROC↑ | wiki FPR↓ | wiki AUROC↑ | Average FPR↓ | Average AUROC↑ |
|---|---|---|---|---|---|---|---|---|---|---|---|---|
| MSP | 96.59 | 46.61 | 86.87 | 70.19 | 95.03 | 47.24 | 94.31 | 46.99 | 95.46 | 54.70 | 93.65 | 53.15 |
| Energy | 96.77 | 44.13 | 85.09 | 68.13 | 94.10 | 51.35 | 96.82 | 46.03 | 97.31 | 29.02 | 94.02 | 47.73 |
| KNN | 90.78 | 66.74 | 84.22 | 73.58 | 95.35 | 57.96 | 90.54 | 53.45 | 93.43 | 43.69 | 90.86 | 59.08 |
| ODIN | 96.74 | 44.69 | 85.65 | 68.95 | 95.06 | 47.36 | 94.27 | 45.20 | 97.88 | 29.91 | 93.92 | 47.22 |
| Mahalanobis | 71.73 | 74.89 | OOM | OOM | 88.60 | 59.57 | 96.03 | 58.50 | 72.33 | 67.95 | 82.17 | 65.23 |
| GKDE | OOT | OOT | OOM | OOM | OOM | OOM | OOM | OOM | OOM | OOM | - | - |
| GPN | OOM | OOM | OOM | OOM | 95.62 | 50.97 | OOM | OOM | OOM | OOM | 95.62 | 50.97 |
| OODGAT | OOM | OOM | OOM | OOM | 92.90 | 59.38 | OOM | OOM | OOM | OOM | 92.90 | 59.38 |
| GNNSafe | 99.49 | 31.99 | 77.86 | 85.66 | 100.00 | 35.30 | 99.92 | 27.35 | 72.63 | 60.32 | 89.98 | 48.12 |
| **GRASP (Ours)** | **2.41** | **98.50** | **39.77** | **93.79** | **73.93** | **81.24** | **75.22** | **72.13** | **58.49** | **77.97** | **49.96** | **84.73** |

(OOD) detection research, which focuses on refining the training strategy to improve graph OOD detection performance. The compared methods include GKDE [92], GPN [64] and OODGAT [63]. While these approaches necessitate a costly re-training procedure, they perform mediocrely across all small-scale datasets and even run out-of-memory on almost all large-scale benchmarks, rendering them impractical for real-world deployment.

**GRASP is performant on challenging heterophily datasets.** As indicated in Table 2 and 3, GNNSafe, which performs well on homophily datasets, experiences significant degradation on the difficult heterophily benchmarks due to its naïve propagation mechanism. In contrast, GRASP maintains optimal performance on these hard scenarios.

## 5.2 A Comprehensive Analysis of GRASP

**Ablation study on augmentation policy.** Recall that the key part of our method GRASP is the augmentation policy that consists in adding edges to the training nodes with top 50% scores of $h(i)$, which corresponds to the nodes on the right side of Figure 5. We ablate the contributions of $h(i)$ by comparison with alternative augmentation approaches that utilize $h(i)$ differently in Table 4, specifically, (1) selecting 50% of training nodes with the lowest $h(i)$ values (left side of Figure 5), (2) randomly selecting 50% of training nodes, corresponding to randomly picking node indices from the x-axis of Figure 5, (3) directly adding edges to $S_{id}$ and $S_{ood}$ within the test set (*i.e.*, TestAug), and (4) a classic graph augmentation method named GAug [91],

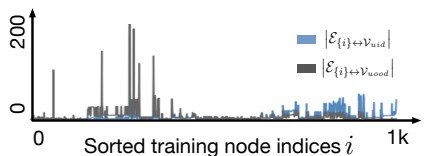

Figure 5: Illustration of the number of edges from each training node $i$ to $S_{id}$ and $S_{ood}$ on Chameleon dataset. The x-axis denotes the training node indices, ordered by $h(i)$ from low to high.

which adds or removes edges based on an edge predictor that disregards $h(i)$ completely. We have the following key observations:

($a$) *Selection by $h(i)$ is effective.* For example, our strategy using the top 50% scores of $h(i)$ outperforms that uses random 50%, which, in turn, outperforms the low 50% way. This is because a higher score of $h(i)$ implies higher edge count towards ID data than to OOD data, which can increase the ratios of intra-edges and improve OOD detection performance after propagation. In contrast, the alternative augmentation method GAug, which does not consider the score $h(i)$ at all, performs even worse than the Low 50% policy.

Table 4: Ablation study on OOD detection performance by different augmentation policy. We report averaged AUROC over 5 independently pre-trained GCN models.

| Strategy | Cora | Amazon | Coauthor | Chameleon | Squirrel | Average |
|---|---|---|---|---|---|---|
| GAug | 64.94 | 74.38 | 91.41 | 63.79 | 47.96 | 68.50 |
| TestAug | 59.24 | 75.78 | 95.00 | 50.58 | 48.64 | 65.85 |
| Low 50% | 88.84 | 95.02 | 97.21 | 54.60 | 53.52 | 77.84 |
| Random 50% | 90.23 | 95.37 | 97.45 | 65.32 | 57.59 | 81.19 |
| Top 50% (**Ours**) | **93.50** | **96.68** | **97.75** | **76.93** | **61.09** | **85.19** |

Table 5: GRASP consistently enhances the OOD detection performance of nodes connected by both inter-edges and intra-edges on datasets characterized by a strong degree of heterophily (datasets highlighted in bold in the table). However, naive propagation tends to compromise the performance of these nodes.

| Datasets | MSP | | MSP+prop | | MSP+**GRASP** | |
|---|---|---|---|---|---|---|
| | intra | inter | intra | inter | intra | inter |
| cora | 48.06 | 46.90 | 60.91 | 48.31 | 78.42 | 50.48 |
| amazon | 66.70 | 57.82 | 81.80 | 68.58 | 89.56 | 78.66 |
| coauthor | 87.72 | 73.30 | 93.88 | 81.37 | 94.85 | 83.21 |
| **chameleon** | 57.48 | 51.22 | 55.63 | 51.02 | 70.79 | 55.19 |
| **squirrel** | 44.76 | 38.45 | 43.74 | 37.04 | 49.89 | 41.69 |
| reddit2 | 64.85 | 54.85 | 65.70 | 54.99 | 96.56 | 93.63 |
| ogbn-product | 31.72 | 37.73 | 37.10 | 43.63 | 67.95 | 64.30 |
| **arxiv-year** | 60.18 | 8.34 | 62.93 | 4.27 | 70.42 | 16.21 |
| **snap-patents** | 61.49 | 5.76 | 64.12 | 0.37 | 67.24 | 9.89 |
| wiki | 56.58 | 48.39 | 60.62 | 52.06 | 69.13 | 61.19 |

(b) *Directly adding edges to $S_{id}$ and $S_{ood}$ within the test set is sub-optimal.* Specifically, employing this method leads to nearly **20**% lower than that achieved with GRASP, which substantiates the notion that "confirmation bias" can adversely affect the graph OOD detection.

Overall, the ablation study suggests that our proposed augmentation policy is crucial to OOD detection performance.

**GRASP can effectively boost performance of challenging nodes connected by inter-edges.** As stated in the introduction 1, the reason OOD score propagation does not always work is the confusion between ID and OOD nodes resulting from propagation along the inter-edges. For example in heterophily datasets, where connected nodes tend to possess different labels, OOD nodes are more likely to appear on the inter-edges. To assess the capability of our proposed augmentation propagation method to address this challenge, we present in Table 5 the accuracy of detecting OOD nodes connected by intra-edges (`intra`) and inter-edges (`inter`) respectively, using the original MSP without any propagation, naive propagation based on MSP (`MSP+prop`), and our proposed augmentation propagation (`MSP+GRASP`) respectively. From the results we can see that naive propagation performs well only on strong homophily datasets, while on strong heterophily datasets (datasets highlighted in bold in the table), its performance is even worse than without propagation, as expected. In contrast, employing our augmentation method still results in substantial performance gain after propagation on these challenging datasets.

**GRASP is compatible with a wide range of OOD scoring methods.** In Table 6, we demonstrate the compatibility of GRASP with various alternative scoring functions. Each method generates OOD scores to form a scoring vector; GRASP is then applied to facilitate score propagation. The use of GRASP markedly surpasses the performance of its non-augmented counterpart across all datasets.

Table 6: GRASP is compatible with different OOD scoring functions. We compare OOD detection methods and the performance after the simple propagation in Equation 1 (denoted by "+ prop") and with GRASP respectively. We report AUROC results that are averaged over 5 independent pre-trained GCN models.

| Method | Cora | Amazon | Coauthor | Chameleon | Squirrel |
|---|---|---|---|---|---|
| MSP | 84.56 | 89.34 | 94.34 | 57.96 | 48.51 |
| MSP+prop | 88.02 | 95.32 | 97.15 | 50.35 | 36.21 |
| MSP+**GRASP** | **93.50** | **96.68** | **97.75** | **76.93** | **61.09** |
| Energy | 85.47 | 90.28 | 95.67 | 59.20 | 45.07 |
| Energy+prop | 87.52 | 96.27 | 95.82 | 50.42 | 36.49 |
| Energy+**GRASP** | **88.34** | **96.35** | **96.64** | **62.04** | **60.66** |
| KNN | 70.94 | 84.71 | 90.13 | 57.90 | 54.68 |
| KNN+prop | 73.70 | 92.36 | 95.47 | 49.76 | 53.99 |
| KNN+**GRASP** | **91.48** | **97.43** | **96.52** | **76.32** | **60.24** |

**Remark on other empirical findings.** We include other empirical findings in Appendix. Specifically, in Table 14, we prove that GRASP also achieves superior performance on the other GNN architectures in the literature. In Figure 7, we demonstrate a strong positive correlation between ratios of intra-edges and the corresponding OOD detection performance, validating the correctness of Theorem 3.2.

We also show in Table 15 that GRASP can increase the ratio of intra-edges after augmentation, consequently boosting OOD performance. This substantiates the correctness of Theorem 4.2. What's more, we conduct a thorough accounting of computation/memory demands compared to baselines on all scale datasets in Table 16 and Table 17 respectively, which underscore the strong practicality of our approach. Lastly, in Appendix D.7, we investigate the other propagation mechanisms in the literature and their impact on graph OOD detection performance emperically.

## 6 Related Work

**Out-of-distribution Detection.** The primary focus within this realm has been on the development of scoring functions for OOD detection. These works can be broadly categorized into two main streams: (1) output-based methods [23, 43, 47, 71, 25, 72, 95, 26, 15, 90, 16, 21], and (2) feature-based methods including the Mahalanobis distance [37, 60, 56] and KNN distance [67]. These methodologies are predominantly applied in domains such as computer vision, where samples are inherently independent of each other. However, these techniques are not designed to adeptly handle data structures like graphs, where samples are inter-connected.

**Out-of-distribution detection for graph data.** Graph anomaly detection has a rich history [12, 74, 89, 75, 48, 14, 50, 32, 46, 51]. In recent years, the OOD detection in graph data introduced fresh challenges, particularly with multi-class classification for in-distribution data, escalating the difficulty in discerning outlier data. Some of the works focus on graph-level OOD detection [42, 49, 4, 76]. For node-level OOD detection, GKDE [92] and GPN [64] apply Bayesian Network models to estimate uncertainties to detect OOD nodes. However, Bayesian-based approaches can encounter impediments such as inaccurate predictions and high computational demands, which limit their broader applicability [83]. GNNSafe [77] emerges as the work employing post hoc energy-based score to perform OOD detection. Given the merits of post hoc methods, our study first provides a comprehensive understanding of the OOD score propagation in Graphs, extending beyond existing knowledge.

**Graph Data Augmentation.** Graph Data Augmentation is a common technique in graph machine learning [20, 6, 58, 29, 93, 91, 33, 53, 13, 3] to improve the node classification performance. Existing methods operate exclusively on in-distribution (ID) data. Furthermore, their test set data also originates from the in-distribution and shares the same classes as the training set. In contrast, our data augmentation is purposefully crafted for OOD detection, supported by the theoretical explanation.

## 7 Conclusions

In this research, we delve into an important yet under-explored challenge in the realm of graph data: Out-of-Distribution (OOD) detection. Recognizing the inadequacies of traditional OOD detection techniques in the context of graph data, our exploration centered on the potential of score propagation as a viable and efficient solution. Our findings reveal the specific conditions under which score propagation will be helpful—in situations where the ratio of intra-edges surpasses that of inter-edges. Motivated by this finding, our edge augmentation strategy selectively adds edges to a specific subset $G$ of the training set, which provably improves post-propagation OOD detection outcomes under certain conditions. Extensive empirical evaluations reinforced the merit of our approach.

## 8 Acknowledgements

This work was supported by the National Natural Science Foundation of China (62441605), and the Starry Night Science Fund of Zhejiang University Shanghai Institute for Advanced Study (SN-ZJUSIAS-0010).

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

# A Proofs

**Theorem A.1.** *(Recap of Theorem 3.2) For any two test ID/OOD node set $S_{id} \subset \mathcal{V}_{uid}, S_{ood} \subset \mathcal{V}_{uood}$ with equal size $N_s$, let the ID-vs-OOD separability $\mathcal{M}_{sep}$ defined on a OOD scoring vector $\hat{\mathbf{g}} \in \mathbb{R}^N$ as*

$$\mathcal{M}_{sep}(\hat{\mathbf{g}}) \triangleq \mathbb{E}_{i \in \mathcal{S}_{id}} \hat{\mathbf{g}}_i - \mathbb{E}_{j \in \mathcal{S}_{ood}} \hat{\mathbf{g}}_j.$$

*If $\mathcal{M}_{sep}(\hat{\mathbf{g}}) > 0$ and $\eta_{intra} - \eta_{inter} > 1/N_s$, for some $\epsilon > 0$ and constant c, we have*

$$\mathbb{P}\left(\mathcal{M}_{sep}(A\hat{\mathbf{g}}) \geq \mathcal{M}_{sep}(\hat{\mathbf{g}}) - \epsilon\right) \geq 1 - exp(-\frac{c\epsilon^2}{\|\hat{\mathbf{g}}\|_2^2}).$$

*Proof.* Without losing the generality, we set the $\hat{\mathbf{g}}_i = 0$, if $i \in S_{ood} \cup S_{id}$, since we only care about the detection results in the given test node set $S_{ood}$ and $S_{id}$.

The $\mathcal{M}_{sep}(\hat{\mathbf{g}})$ can be re-written as

$$\mathcal{M}_{sep}(\hat{\mathbf{g}}) = \hat{\mathbf{g}}^\top (\mathbf{e}_{S_{id}} - \mathbf{e}_{S_{ood}}).$$

Then we have

$$\mathcal{M}_{sep}(A\hat{\mathbf{g}}) = \hat{\mathbf{g}}^\top A(\mathbf{e}_{S_{id}} - \mathbf{e}_{S_{ood}})$$

According to General Hoeffding's inequality (Theorem 2.6.3) in [70], we know that

$$\mathbb{P}\left(\mathbb{E}[\hat{\mathbf{g}}^\top A(\mathbf{e}_{S_{id}} - \mathbf{e}_{S_{ood}})] - \hat{\mathbf{g}}^\top A(\mathbf{e}_{S_{id}} - \mathbf{e}_{S_{ood}}) \leq \epsilon\right) \geq 1 - \exp(-\frac{c\epsilon^2}{\|\hat{\mathbf{g}}\|_2^2}),$$

where $c$ is some constant value.

Since $\hat{\mathbf{g}}_i = 0$, if $i \in \mathcal{V}_l$,

$$\begin{aligned}
\mathbb{E}[\hat{\mathbf{g}}^\top A(\mathbf{e}_{S_{id}} - \mathbf{e}_{S_{ood}})] &= \hat{\mathbf{g}}^\top \mathbb{E}[A](\mathbf{e}_{S_{id}} - \mathbf{e}_{S_{ood}}) \\
&= \hat{\mathbf{g}}^\top N_s(\eta_{intra} - \eta_{inter})(\mathbf{e}_{S_{id}} - \mathbf{e}_{S_{ood}}) \\
&> \hat{\mathbf{g}}^\top (\mathbf{e}_{S_{id}} - \mathbf{e}_{S_{ood}})
\end{aligned}$$

Combining together, we have

$$\mathbb{P}\left(\hat{\mathbf{g}}^\top A(\mathbf{e}_{S_{id}} - \mathbf{e}_{S_{ood}}) \geq \hat{\mathbf{g}}^\top (\mathbf{e}_{S_{id}} - \mathbf{e}_{S_{ood}}) - \epsilon\right) \geq 1 - \exp(-\frac{c\epsilon^2}{\|\hat{\mathbf{g}}\|_2^2})$$

$\square$

**Theorem A.2.** *(Recap of Theorem 4.2) For any two test ID/OOD node set $S_{id} \subset \mathcal{V}_{uid}, S_{ood} \subset \mathcal{V}_{uood}$ with equal size $N_s$, let the ID-vs-OOD separability $\mathcal{M}_{sep}$ defined on a non-negative OOD scoring vector $\hat{\mathbf{g}} \in \mathbb{R}^N$ as*

$$\mathcal{M}_{sep}(\hat{\mathbf{g}}) \triangleq \mathbb{E}_{i \in \mathcal{S}_{id}} \hat{\mathbf{g}}_i - \mathbb{E}_{j \in \mathcal{S}_{ood}} \hat{\mathbf{g}}_j.$$

*Let $\mathcal{E}_{S \leftrightarrow S'} \subset \mathcal{E}$ to denote the edge set of edges between two node sets $S$ and $S'$, where $S, S' \subset \mathcal{V}$. If we can find a node set $G \subset \mathcal{V}_l$ such that $|\mathcal{E}_{G \leftrightarrow S_{id}}| > |\mathcal{E}_{G \leftrightarrow S_{ood}}|$, we have*

$$\mathcal{M}_{sep}((A + \delta E)^2 \hat{\mathbf{g}}) > \mathcal{M}_{sep}(A^2 \hat{\mathbf{g}}),$$

*where $E = \mathbf{e}_G \mathbf{e}_G^\top$ and $\delta > 0$.*

*Proof.* The $\mathcal{M}_{sep}(\hat{\mathbf{g}})$ can be re-written as

$$\mathcal{M}_{sep}(\hat{\mathbf{g}}) = \frac{1}{N_s} \hat{\mathbf{g}}^\top (\mathbf{e}_{S_{id}} - \mathbf{e}_{S_{ood}}).$$

We can then directly derive the proof by expanding

$$\mathcal{M}_{sep}((A+\delta E)^2\hat{\mathbf{g}}) - \mathcal{M}_{sep}(A^2\hat{\mathbf{g}}) = \frac{1}{N_s}(\mathbf{e}_{S_{id}} - \mathbf{e}_{S_{ood}})^\top \left((A+\delta E)^2\hat{\mathbf{g}} - A^2\hat{\mathbf{g}}\right)$$

$$= \frac{1}{N_s}(\mathbf{e}_{S_{id}} - \mathbf{e}_{S_{ood}})^\top \left(\delta(AE + EA)\hat{\mathbf{g}} + \delta^2 \mathbf{e}_G \mathbf{e}_G^\top \mathbf{e}_G \mathbf{e}_G^\top \hat{\mathbf{g}}\right)$$

$$= \frac{\delta}{N_s}(\mathbf{e}_{S_{id}} - \mathbf{e}_{S_{ood}})^\top \left(A\mathbf{e}_G\mathbf{e}_G^\top\hat{\mathbf{g}} + \mathbf{e}_G\mathbf{e}_G^\top A\hat{\mathbf{g}}\right)$$

$$= \frac{\delta}{N_s}(\mathbf{e}_{S_{id}} - \mathbf{e}_{S_{ood}})^\top A\mathbf{e}_G\mathbf{e}_G^\top\hat{\mathbf{g}}$$

$$= \frac{\delta}{N_s}(\mathbf{e}_G^\top\hat{\mathbf{g}})(\mathbf{e}_{S_{id}}^\top A\mathbf{e}_G - \mathbf{e}_{S_{ood}}^\top A\mathbf{e}_G)$$

$$= \frac{\delta(\mathbf{e}_G^\top\hat{\mathbf{g}})}{|G|N_s}(|\mathcal{E}_{G\leftrightarrow S_{id}}| - |\mathcal{E}_{G\leftrightarrow S_{ood}}|)$$

$$> 0,$$

where the second and the third equation are derived by the fact that $G \subset \mathcal{V}_l$ and then we have $\mathbf{e}_{S_{id}}^\top\mathbf{e}_G = 0$ and $\mathbf{e}_{S_{ood}}^\top\mathbf{e}_G = 0$.

$\square$

## B   Details of Baselines

For the reader's convenience, we summarize in detail a few common techniques for defining OOD scores that measure the degree of ID-ness on a given input. By convention, a higher (lower) score is indicative of being in-distribution (out-of-distribution).

**MSP [23]** This method proposes to use the maximum softmax score as the OOD score. For each node $i$, we use $F_{OODD}(i) = \max_{c\in[C]} f_c(i)$ as the OOD score.

**ODIN [43]** This method improves OOD detection with temperature scaling and input perturbation. In all experiments, we set the temperature scaling parameter $T = 1000$. For graph neural network, we found the input perturbation does not further improve the OOD detection performance and hence we set $\epsilon = 0$.

**Mahalanobis [37]** This method uses multivariate Gaussian distributions to model class-conditional distributions of softmax neural classifiers and uses Mahalanobis distance-based scores for OOD detection. The mean $\mu_c$ of each multivariate Gaussian distribution with class $c$ and a tied covariance $\mathbf{\Sigma}$ are estimated based on training samples. We define the confidence score $M(\mathbf{x})$ using the Mahalanobis distance between test sample $\mathbf{x}$ and the closest class-conditional Gaussian distribution.

**Energy [47]** This method proposes using energy score for OOD detection. The energy function maps the logit outputs to a scalar $E(\mathbf{x}_i; f) \in \mathbb{R}$, which is relatively lower for ID data. Note that [47] used the *negative energy score* for OOD detection, in order to align with the convention that $S(\mathbf{x})$ is higher (lower) for ID (OOD) data.

**KNN [67]** This method uses the $k$-th nearest neighbor distance between a test graph node and the training set as the OOD score. We use $k = 10$ for all experiments in this paper.

## C   Dataset Details

We adopt ten publicly available benchmarks used for graph learning, covering diverse domains, scales, and structures (homophily or heterophily). For five homophily datasets `Cora`, `Amazon-Photo`, `Coauthor-CS`, `Reddit2` and `ogbn-products`, we use the data loader provided by the Pytorch Geometric package [3]. For the remaining five heterophily datasets, we directly use the pickle file or download from the given hyperlinks proposed by [44].

`Cora` [61] is a 7-class citation network comprising 2,708 nodes, 5,429 edges and 1,433 features. In this network, each node represents a published paper, each edge signifies a citation relationship, and the label class is each paper's topic, which is the goal to predict.

---

[3]https://pytorch-geometric.readthedocs.io/en/latest/modules/datasets.html

`Amazon-Photo` [52] is an 8-class item co-purchasing network on Amazon, which contains 7,650 nodes, 238,162 edges and 745 features. In this network, each node denotes a product, each edge indicates that two linked products are frequently purchased together, and the node label denotes the category of the product.

`Coauthor-CS` [62] is a 15-class coauthor network of computer science, which contains 18,333 nodes, 163,788 edges and 6,805 features. In this network, nodes denote authors and there is an edge between two authors if co-authored a paper. And the label represents the study field for the authors.

`Reddit2` [88] is a large homophily post network with 41 classes, where nodes are posts based on user comments and the task is to predict communities of online posts based on user comments.

`ogbn-products` [24] is a super large homophily undirected and unweighted graph with 47 classes, representing an Amazon product co-purchasing network. Nodes represent products sold in Amazon, and edges between two products indicate that the products are purchased together. The task is to predict the category of a product in a multi-class classification setup, where the 47 top-level categories are used for target labels.

`Chameleon` and `Squirrel` [59] are two Wikipedia networks with 5 classes, where nodes represent web pages and edges represent hyperlinks between them. Node features represent several informative nouns in the Wikipedia pages and the task is to predict the average daily traffic of the web page [17].

`arXiv-year` [24] is the ogbn-arXiv network with different labels and is altered to be heterophily, in which the class labels are set to be the year that the paper is posted, instead of subject area in the original paper. The nodes are arXiv papers, and directed edges connect a paper to other papers that it cites. The node features are averaged word2vec token features of both the title and abstract of the paper. The five classes are chosen by partitioning the posting dates so that class ratios are approximately balanced [44].

`snap-patents` [39, 38] is a big dataset of utility patents in the US. Each node represents a patent and edges connect patents that cite each other. Node features are derived from patent metadata [44]. Like `arXiv-year`, this dataset is changed to set the task to predict the time at which a patent was granted, which is also five classes.

`wiki` [44] is a super big dataset of Wikipedia articles, which are crawled and cleaned from the internet. Nodes represent pages and edges represent links between them. Node features are derived from the average GloVe embeddings [54] of the titles and abstracts and labels indicate total page views over a 60-day period, categorized into five classes based on quintiles.

The complete information and statistics of all these datasets aforementioned are summarized in Table 7.

Table 7: Statistics of all the graph datasets. # C is the total number of distinct node classes.

| Dataset | # Nodes | # Edges | # Features | # C | Domain | Homoph | OOD Class | ID Class |
|---|---|---|---|---|---|---|---|---|
| Cora | 2,708 | 5,429 | 1,433 | 7 | citation | ✓ | $\{0, \cdots, 3\}$ | $\{4, 5, 6\}$ |
| Amazon-Photo | 7,650 | 238,162 | 745 | 8 | product | ✓ | $\{0, \cdots, 4\}$ | $\{5, 6, 7\}$ |
| Coauthor-CS | 18,333 | 163,788 | 6,805 | 15 | citation | ✓ | $\{0, \cdots, 3\}$ | $\{4, \cdots, 14\}$ |
| Reddit2 | 232,965 | 23,213,838 | 602 | 41 | post | ✓ | $\{0, \cdots, 10\}$ | $\{11, \cdots, 40\}$ |
| ogbn-products | 2,449,029 | 61,859,140 | 100 | 47 | product | ✓ | $\{0, \cdots, 11\}$ | $\{12, \cdots, 46\}$ |
| Chameleon | 2,277 | 31,421 | 2,325 | 5 | Wikipedia | ✗ | $\{0, 1\}$ | $\{2, 3, 4\}$ |
| Squirrel | 5,201 | 198,493 | 2,089 | 5 | Wikipedia | ✗ | $\{0, 1\}$ | $\{2, 3, 4\}$ |
| arXiv-year | 169,343 | 1,166,243 | 128 | 5 | citation | ✗ | $\{0, 1\}$ | $\{2, 3, 4\}$ |
| snap-patents | 2,923,922 | 13,975,788 | 269 | 5 | patent | ✗ | $\{0, 1\}$ | $\{2, 3, 4\}$ |
| wiki | 1,925,342 | 303,434,860 | 600 | 5 | Wikipedia | ✗ | $\{0, 1\}$ | $\{2, 3, 4\}$ |

# D  Additional Experiments

## D.1  Graph ID Classification Details

In this section, we provide a detailed report on the ID ACC results of nine popular GNN pretrained backbones used in our paper, including GCN [34], GAT [69], GCNJK [81], GATJK [81], APPNP [35], MixHop [1], GPR-GNN [10], GCNII [7], and $H_2$GCN [94]. The results of all the nine architectures on five common benchmarks and architecture GCN on five large-scale benchmarks are presented in Table 8 and Table 9 respectively. In addition, we present the ID ACC results of three training-based methods on the ten benchmarks. It is important to note that these training-based methods only yield results on five small-scale datasets and the moderately sized arxiv-year dataset. They fail to produce results on the remaining four large datasets, as shown in Tables 10 and Table 11.

Table 8: ID ACCs of six pre-trained methods on common benchmarks. For each pre-trained method, we take the average values that are percentages over 5 independently trained backbones.

| Backbone | Cora | Amazon | Coauthor | Chameleon | Squirrel |
|---|---|---|---|---|---|
| GCN | $93.89 \pm 1.31$ | $96.72 \pm 0.76$ | $96.21 \pm 0.61$ | $71.20 \pm 2.07$ | $72.32 \pm 2.02$ |
| GAT | $93.89 \pm 1.64$ | $96.44 \pm 0.92$ | $95.67 \pm 0.35$ | $73.43 \pm 3.42$ | $75.88 \pm 1.73$ |
| GCNJK | $92.92 \pm 2.03$ | $96.88 \pm 0.56$ | $95.93 \pm 0.32$ | $71.03 \pm 2.41$ | $72.22 \pm 0.71$ |
| GATJK | $93.27 \pm 1.91$ | $96.31 \pm 0.38$ | $95.78 \pm 0.26$ | $74.55 \pm 1.95$ | $76.47 \pm 1.45$ |
| GCNII | $94.07 \pm 1.28$ | $96.62 \pm 0.57$ | $96.97 \pm 0.37$ | $73.08 \pm 1.78$ | $72.57 \pm 1.92$ |
| $H_2$GCN | $94.07 \pm 1.84$ | $96.26 \pm 0.74$ | $94.57 \pm 0.43$ | $71.85 \pm 2.21$ | $74.73 \pm 1.11$ |
| APPNP | $91.44 \pm 1.48$ | $95.63 \pm 0.19$ | $95.79 \pm 0.28$ | $54.77 \pm 1.66$ | $42.87 \pm 2.07$ |
| MixHop | $86.05 \pm 4.01$ | $93.47 \pm 1.31$ | $93.80 \pm 0.28$ | $59.48 \pm 3.15$ | $53.18 \pm 0.69$ |
| GPR-GNN | $91.02 \pm 1.76$ | $95.35 \pm 0.14$ | $96.04 \pm 0.16$ | $46.24 \pm 10.26$ | $45.73 \pm 0.95$ |

Table 9: ID ACCs of pre-trained GCN models on five large-scale datasets. We report the average values that are percentages over 5 independently trained backbones.

| Backbone | reddit2 | ogbn-products | arxiv-year | snap-patents | wiki |
|---|---|---|---|---|---|
| GCN | $51.36 \pm 0.35$ | $74.39 \pm 0.09$ | $56.67 \pm 0.33$ | $62.48 \pm 0.10$ | $54.89 \pm 0.16$ |

Table 10: ID ACCs of three training-based ood detection methods on five small-scale datasets. For each method, we take the average values that are percentages over 5 independently runs.

| Method | Cora | Amazon | Coauthor | Chameleon | Squirrel |
|---|---|---|---|---|---|
| GKDE | $91.40 \pm 1.39$ | $91.30 \pm 2.30$ | $96.14 \pm 0.50$ | $65.22 \pm 1.67$ | $53.29 \pm 2.97$ |
| GPN | $91.49 \pm 1.51$ | $93.96 \pm 2.39$ | $89.83 \pm 4.25$ | $62.40 \pm 1.40$ | $46.27 \pm 5.37$ |
| OODGAT | $76.02 \pm 7.77$ | $61.88 \pm 1.19$ | $31.71 \pm 1.06$ | $61.00 \pm 2.84$ | $50.88 \pm 2.35$ |

Table 11: ID ACCs of three training-based methods on five large-scale datasets, where OOM means Out-Of-Memory and OOT denotes that no results have been got after running over 48 hours for each run. All the training-based methods only have results on the moderately sized arXiv-year dataset. For each method, we take the average values that are percentages over 5 independently runs.

| Method | reddit2 | ogbn-products | arxiv-year | snap-patents | wiki |
|---|---|---|---|---|---|
| GKDE | OOM | OOM | OOT | OOM | OOM |
| GPN | OOM | OOM | $43.07 \pm 2.02$ | OOM | OOM |
| OODGAT | OOM | OOM | $43.43 \pm 0.36$ | OOM | OOM |

## D.2 Detailed Main Graph OOD Detection Results

Table 12: **Detailed main results on common benchmarks.** Comparison with competitive out-of-distribution detection methods on pre-trained GCN. We take the average values with standard errors that are percentages over 5 independently trained backbones. ↑ (↓) indicates larger (smaller) values are better.

| Method | Cora | | Amazon | | Coauthor | | Chameleon | | Squirrel | |
|---|---|---|---|---|---|---|---|---|---|---|
| | FPR↓ | AUROC↑ | FPR↓ | AUROC↑ | FPR↓ | AUROC↑ | FPR↓ | AUROC↑ | FPR↓ | AUROC↑ |
| MSP | 70.86 ± 15.88 | 84.56 ± 5.39 | 49.26 ± 10.51 | 89.34 ± 3.49 | 28.82 ± 1.94 | 94.34 ± 0.41 | 85.70 ± 7.09 | 57.96 ± 3.31 | 94.68 ± 1.01 | 48.51 ± 0.46 |
| Energy | 67.54 ± 22.98 | 85.47 ± 4.98 | 42.13 ± 9.96 | 90.28 ± 3.42 | 20.29 ± 1.49 | 95.67 ± 0.25 | 88.06 ± 7.50 | 59.20 ± 4.31 | 93.98 ± 1.42 | 45.07 ± 1.68 |
| KNN | 90.20 ± 4.35 | 70.94 ± 5.62 | 65.19 ± 7.26 | 84.71 ± 3.28 | 51.24 ± 1.83 | 90.13 ± 0.50 | 93.38 ± 5.48 | 57.90 ± 6.48 | 94.72 ± 2.84 | 54.68 ± 2.25 |
| ODIN | 68.41 ± 18.48 | 84.98 ± 5.59 | 44.06 ± 10.69 | 89.90 ± 3.65 | 22.59 ± 1.76 | 95.27 ± 0.33 | 85.31 ± 7.64 | 57.94 ± 3.75 | 94.17 ± 0.44 | 44.08 ± 0.35 |
| Mahalanobis | 69.68 ± 14.60 | 85.48 ± 1.69 | 96.49 ± 5.96 | 75.58 ± 7.97 | 85.71 ± 1.82 | 84.98 ± 0.58 | 95.55 ± 2.36 | 53.19 ± 4.30 | 94.90 ± 0.51 | 54.99 ± 0.70 |
| GKDE | 63.71 ± 14.36 | 86.27 ± 2.69 | 81.29 ± 3.36 | 77.26 ± 5.54 | 25.48 ± 1.48 | 95.13 ± 0.29 | 92.93 ± 4.89 | 50.14 ± 5.50 | 96.71 ± 0.67 | 49.38 ± 3.58 |
| GPN | 58.45 ± 31.98 | 82.93 ± 11.20 | 72.95 ± 19.77 | 82.63 ± 5.87 | 34.11 ± 22.46 | 93.82 ± 2.63 | 82.25 ± 6.55 | 68.20 ± 6.70 | 95.58 ± 1.65 | 48.38 ± 4.43 |
| OODGAT | 94.59 ± 6.38 | 53.63 ± 5.13 | 71.34 ± 15.34 | 66.95 ± 16.02 | 96.53 ± 3.39 | 52.18 ± 8.26 | 94.43 ± 3.43 | 59.67 ± 6.37 | 95.27 ± 1.00 | 46.13 ± 3.10 |
| GNNSafe | 54.71 ± 31.41 | 87.52 ± 6.16 | 22.39 ± 4.90 | 96.27 ± 0.31 | 16.64 ± 1.90 | 95.82 ± 0.28 | 100.00 ± 0.00 | 50.42 ± 0.65 | 100.00 ± 0.00 | 35.88 ± 0.24 |
| NODESafe | 45.73 ± 21.86 | 89.70 ± 4.60 | 63.25 ± 23.15 | 79.03 ± 13.00 | 39.06 ± 25.23 | 87.45 ± 13.31 | 100.00 ± 0.00 | 50.34 ± 0.58 | 100.00 ± 0.00 | 36.18 ± 0.23 |
| fDBD | 81.56 ± 10.45 | 56.77 ± 12.97 | 51.87 ± 13.89 | 73.31 ± 7.86 | 59.68 ± 3.50 | 63.10 ± 1.01 | 89.85 ± 11.80 | 50.85 ± 14.89 | 94.78 ± 1.30 | 53.17 ± 3.24 |
| GRASP | 29.70 ± 12.25 | 93.50 ± 1.65 | 14.38 ± 6.63 | 96.68 ± 0.28 | 7.84 ± 0.58 | 97.75 ± 0.18 | 66.88 ± 6.48 | 76.93 ± 4.18 | 85.59 ± 3.61 | 61.09 ± 1.49 |

Table 13: **Detailed main results on large-scale benchmarks.** Comparison with competitive out-of-distribution detection methods on pre-trained method GCN. We take the average values with standard errors that are percentages over 5 independently trained backbones. OOM means Out-Of-Memory and OOT denotes that no results have been got after running over 48 hours for each run. ↑ (↓) indicates larger (smaller) values are better.

| Method | reddit2 | | ogbn-products | | arxiv-year | | snap-patents | | wiki | |
|---|---|---|---|---|---|---|---|---|---|---|
| | FPR↓ | AUROC↑ | FPR↓ | AUROC↑ | FPR↓ | AUROC↑ | FPR↓ | AUROC↑ | FPR↓ | AUROC↑ |
| MSP | 96.59 ± 0.14 | 46.61 ± 0.66 | 86.87 ± 0.35 | 70.19 ± 0.92 | 95.03 ± 1.46 | 47.24 ± 3.70 | 94.31 ± 0.30 | 46.99 ± 0.83 | 95.46 ± 0.32 | 54.70 ± 0.68 |
| Energy | 96.77 ± 0.03 | 44.13 ± 0.14 | 85.09 ± 0.45 | 68.13 ± 0.38 | 94.10 ± 2.76 | 51.35 ± 5.91 | 96.82 ± 1.07 | 46.03 ± 4.59 | 97.31 ± 1.54 | 29.02 ± 2.78 |
| KNN | 90.78 ± 0.75 | 66.74 ± 0.55 | 84.22 ± 2.00 | 73.58 ± 1.21 | 95.35 ± 0.92 | 57.96 ± 2.19 | 90.54 ± 1.09 | 53.45 ± 0.93 | 93.43 ± 2.57 | 43.69 ± 4.83 |
| ODIN | 96.74 ± 0.07 | 44.69 ± 0.24 | 85.65 ± 0.31 | 68.95 ± 0.52 | 95.06 ± 1.47 | 47.36 ± 3.46 | 94.27 ± 0.30 | 45.20 ± 0.87 | 97.88 ± 0.18 | 29.91 ± 0.47 |
| Mahalanobis | 71.73 ± 1.55 | 74.89 ± 1.01 | OOM | - | 88.60 ± 1.27 | 59.57 ± 1.27 | 96.03 ± 0.22 | 58.50 ± 0.81 | 72.33 ± 2.15 | 67.95 ± 1.56 |
| GKDE | OOT | - | OOM | - | OOM | - | OOM | - | OOM | - |
| GPN | OOM | - | OOM | - | 95.62 ± 3.29 | 50.97 ± 14.98 | OOM | - | OOM | - |
| OODGAT | OOM | - | OOM | - | 92.90 ± 0.94 | 59.38 ± 3.44 | OOM | - | OOM | - |
| GNNSafe | 99.49 ± 0.07 | 31.99 ± 0.26 | 77.86 ± 1.09 | 85.66 ± 1.16 | 100.00 ± 0.00 | 35.30 ± 0.06 | 99.92 ± 0.18 | 27.35 ± 0.18 | 72.63 ± 2.05 | 60.32 ± 4.51 |
| NODESafe | 86.47 ± 10.45 | 47.00 ± 9.33 | OOT | - | 100.00 ± 0.00 | 35.32 ± 0.04 | 100.00 ± 0.00 | 27.27 ± 0.01 | OOT | - |
| fDBD | 89.78 ± 0.73 | 55.72 ± 0.88 | 83.41 ± 0.53 | 66.67 ± 1.15 | 95.82 ± 2.23 | 48.87 ± 9.73 | 94.99 ± 0.20 | 43.27 ± 2.50 | 96.53 ± 0.55 | 59.32 ± 1.23 |
| GRASP | 2.41 ± 0.09 | 98.50 ± 0.02 | 39.77 ± 1.25 | 93.79 ± 0.24 | 73.93 ± 0.60 | 81.24 ± 0.39 | 75.22 ± 0.09 | 72.13 ± 0.06 | 58.49 ± 1.07 | 77.97 ± 1.38 |

## D.3 Graph OOD Detection Results on Various Backbones

GRASP can seamlessly be applied to various GNN backbones. In this section, we report the OOD detection performance of all baselines on various popular architectures in Table 14

Table 14: Results of various GNN pretrained backbones on common benchmarks.

| Pre-trained Backbone | OOD Detection Method | Cora FPR↓ | Cora AUROC↑ | Amazon FPR↓ | Amazon AUROC↑ | Coauthor FPR↓ | Coauthor AUROC↑ | Chameleon FPR↓ | Chameleon AUROC↑ | Squirrel FPR↓ | Squirrel AUROC↑ |
|---|---|---|---|---|---|---|---|---|---|---|---|
| GCN | MSP | 70.86 | 84.56 | 49.26 | 89.34 | 28.82 | 94.34 | 85.70 | 57.96 | 94.68 | 48.51 |
| | Energy | 67.54 | 85.47 | 42.13 | 90.28 | 20.29 | 95.67 | 88.06 | 59.20 | 93.98 | 45.07 |
| | KNN | 90.20 | 70.94 | 65.19 | 84.71 | 51.24 | 90.13 | 93.38 | 57.90 | 94.72 | 54.68 |
| | ODIN | 68.41 | 84.98 | 44.06 | 89.90 | 22.59 | 95.27 | 85.31 | 57.94 | 94.17 | 44.08 |
| | Mahalanobis | 69.68 | 85.48 | 96.49 | 75.58 | 85.71 | 84.98 | 95.55 | 53.19 | 94.90 | 54.99 |
| | GNNSafe | 54.71 | 87.52 | 22.39 | 96.27 | 16.64 | 95.82 | 100.00 | 50.42 | 100.00 | 35.88 |
| | **GRASP (ours)** | **29.70** | **93.50** | **14.38** | **96.68** | **7.84** | **97.75** | **66.88** | **76.93** | **85.59** | **61.09** |
| GAT | MSP | 55.33 | 88.82 | 29.88 | 94.39 | 28.15 | 94.26 | 91.27 | 61.94 | 95.21 | 47.50 |
| | Energy | 80.71 | 79.16 | 26.48 | 95.24 | 20.96 | 95.65 | 92.71 | 61.11 | 96.47 | 45.69 |
| | KNN | 71.14 | 81.28 | 46.42 | 90.74 | 42.51 | 91.51 | 89.02 | 61.13 | 95.37 | 53.16 |
| | ODIN | 55.27 | 89.06 | 26.92 | 94.89 | 24.61 | 94.95 | 90.83 | 62.89 | 96.11 | 45.68 |
| | Mahalanobis | 67.92 | 86.37 | 14.28 | 95.80 | 26.27 | 94.46 | 95.35 | 50.65 | 91.36 | 57.67 |
| | GNNSafe | 58.97 | 85.64 | 29.12 | 93.16 | 25.41 | 93.91 | 100.00 | 50.39 | 100.00 | 36.21 |
| | **GRASP (ours)** | **22.76** | **94.28** | **14.21** | **96.79** | **8.59** | **97.51** | **70.15** | **73.40** | **85.84** | **61.18** |
| GCNJK | MSP | 81.33 | 80.40 | 32.45 | 94.64 | 26.43 | 94.44 | 86.42 | 68.19 | 94.93 | 51.83 |
| | Energy | 96.56 | 70.16 | 40.90 | 93.80 | 18.75 | 95.75 | 91.92 | 65.16 | 95.36 | 49.68 |
| | KNN | 90.98 | 73.81 | 64.47 | 85.18 | 50.95 | 89.98 | 94.45 | 59.04 | 94.64 | 53.49 |
| | ODIN | 81.04 | 80.68 | 28.35 | 95.13 | 21.12 | 95.41 | 86.03 | 68.58 | 95.04 | 50.64 |
| | Mahalanobis | 60.84 | 86.20 | 61.61 | 87.11 | 83.04 | 87.34 | 87.23 | 66.61 | 91.52 | 57.24 |
| | GNNSafe | 65.01 | 83.11 | 22.41 | 96.28 | 13.27 | 96.47 | 100.00 | 50.40 | 100.00 | 36.21 |
| | **GRASP (ours)** | **29.69** | **92.98** | **12.66** | **96.86** | **8.03** | **97.74** | **59.61** | **75.78** | **86.02** | **60.70** |
| GATJK | MSP | 69.56 | 84.51 | 47.21 | 91.32 | 24.66 | 95.37 | 94.39 | 55.43 | 94.67 | 50.98 |
| | Energy | 62.27 | 85.75 | 34.75 | 92.89 | 17.23 | 96.38 | 91.11 | 59.01 | 95.61 | 48.76 |
| | KNN | 82.54 | 74.32 | 70.98 | 83.48 | 38.95 | 92.56 | 92.21 | 61.14 | 95.20 | 54.32 |
| | ODIN | 64.25 | 85.21 | 39.29 | 92.19 | 18.16 | 96.30 | 93.56 | 56.10 | 95.24 | 48.62 |
| | Mahalanobis | 79.60 | 79.33 | 52.79 | 88.53 | 34.60 | 93.68 | 91.59 | 52.38 | 91.52 | 56.19 |
| | GNNSafe | 44.43 | 90.01 | 22.46 | 95.45 | 17.54 | 95.32 | 100.00 | 50.39 | 100.00 | 36.15 |
| | **GRASP (ours)** | **29.04** | **92.57** | **14.78** | **96.70** | **8.32** | **97.70** | **78.65** | **71.09** | **85.88** | **61.17** |
| APPNP | MSP | 59.37 | 89.01 | 64.64 | 86.51 | 18.38 | 96.45 | 94.24 | 48.87 | 94.41 | 50.91 |
| | Energy | 81.82 | 81.21 | 62.87 | 84.36 | 14.57 | 97.01 | 90.55 | 55.75 | 90.91 | 53.04 |
| | KNN | 75.33 | 81.21 | 49.55 | 89.76 | 38.44 | 91.71 | 92.14 | 54.19 | 94.12 | 53.14 |
| | ODIN | 56.72 | 89.47 | 60.67 | 86.76 | 15.02 | 96.98 | 94.63 | 50.71 | 94.41 | 50.60 |
| | Mahalanobis | 73.64 | 86.02 | 98.75 | 62.13 | 30.20 | 93.91 | 92.38 | 58.15 | 93.29 | 56.65 |
| | GNNSafe | 59.70 | 85.45 | 19.26 | 95.08 | 12.10 | 96.60 | 100.00 | 50.45 | 100.00 | 36.24 |
| | **GRASP (ours)** | **26.45** | **94.16** | **5.69** | **97.11** | **8.69** | **97.59** | **83.41** | **63.02** | **86.42** | **60.76** |
| H$_2$GCN | MSP | 67.00 | 86.50 | 59.23 | 86.88 | 99.37 | 40.35 | 91.00 | 62.79 | 94.34 | 57.21 |
| | Energy | 68.06 | 86.84 | 57.05 | 86.21 | 97.85 | 51.65 | 92.66 | 63.24 | 96.75 | 53.18 |
| | KNN | 80.00 | 79.68 | 63.85 | 80.54 | 60.66 | 77.25 | 95.13 | 56.89 | 95.62 | 57.45 |
| | ODIN | 65.21 | 87.10 | 56.25 | 86.97 | 99.43 | 41.58 | 91.07 | 63.52 | 95.08 | 55.69 |
| | Mahalanobis | 81.67 | 80.55 | 86.26 | 77.33 | 97.92 | 61.02 | 97.62 | 58.29 | 96.36 | 53.54 |
| | GNNSafe | 43.97 | 88.83 | 33.40 | 90.87 | 93.00 | 43.23 | 100.00 | 50.35 | 100.00 | 36.26 |
| | **GRASP (ours)** | **33.54** | **92.63** | **16.57** | **96.48** | **14.23** | **96.08** | **66.38** | **74.72** | **86.04** | **60.83** |
| MixHop | MSP | 83.94 | 78.60 | 53.56 | 90.97 | 48.66 | 90.91 | 92.95 | 56.77 | 95.60 | 49.07 |
| | Energy | 83.67 | 77.15 | 57.04 | 89.28 | 28.49 | 94.67 | 94.10 | 57.21 | 95.61 | 48.87 |
| | KNN | 93.36 | 69.93 | 65.41 | 86.45 | 62.40 | 85.91 | 89.52 | 57.64 | 93.44 | 54.00 |
| | ODIN | 83.14 | 79.10 | 50.00 | 91.25 | 41.39 | 92.65 | 93.45 | 56.48 | 95.68 | 47.58 |
| | Mahalanobis | 82.35 | 80.04 | 90.05 | 81.85 | 47.41 | 91.67 | 93.93 | 56.30 | 91.33 | 56.56 |
| | GNNSafe | 66.86 | 83.77 | 39.72 | 93.54 | 33.83 | 92.46 | 100.00 | 50.35 | 100.00 | 36.42 |
| | **GRASP (ours)** | **32.11** | **92.77** | **10.07** | **96.99** | **9.41** | **97.31** | **76.92** | **66.12** | **85.92** | **60.69** |
| GPR-GNN | MSP | 64.90 | 87.44 | 62.84 | 87.66 | 23.96 | 95.64 | 96.09 | 47.65 | 95.78 | 44.62 |
| | Energy | 72.85 | 83.86 | 64.23 | 85.28 | 16.42 | 96.50 | 93.78 | 49.09 | 95.16 | 42.63 |
| | KNN | 74.24 | 81.46 | 48.47 | 90.48 | 38.83 | 92.31 | 94.39 | 55.31 | 94.18 | 51.74 |
| | ODIN | 62.58 | 88.13 | 55.49 | 88.41 | 17.24 | 96.51 | 96.16 | 47.50 | 95.51 | 42.32 |
| | Mahalanobis | 79.56 | 84.53 | 97.25 | 69.75 | 49.93 | 91.56 | 87.01 | 55.95 | 87.24 | 61.10 |
| | GNNSafe | 51.65 | 85.91 | 13.63 | 96.46 | 14.73 | 95.96 | 100.00 | 50.32 | 100.00 | 36.25 |
| | **GRASP (ours)** | **26.71** | **94.02** | **5.30** | **97.14** | **8.28** | **97.70** | **76.53** | **72.43** | **85.40** | **61.33** |
| GCNII | MSP | 72.85 | 83.02 | 51.72 | 88.13 | 23.18 | 95.21 | 96.03 | 55.46 | 94.13 | 49.46 |
| | Energy | 83.15 | 75.24 | 48.28 | 88.78 | 17.72 | 96.03 | 95.87 | 56.75 | 94.61 | 48.63 |
| | KNN | 83.99 | 76.02 | 59.25 | 86.74 | 36.05 | 93.43 | 94.72 | 52.86 | 94.65 | 53.47 |
| | ODIN | 71.49 | 83.31 | 49.44 | 88.35 | 19.44 | 95.75 | 95.61 | 56.63 | 94.73 | 48.34 |
| | Mahalanobis | 73.90 | 82.01 | 77.63 | 80.87 | 44.01 | 92.63 | 96.68 | 46.57 | 91.66 | 53.62 |
| | GNNSafe | 66.70 | 83.12 | 27.08 | 93.13 | 17.87 | 94.47 | 100.00 | 50.35 | 100.00 | 36.32 |
| | **GRASP (ours)** | **27.92** | **93.51** | **23.53** | **93.72** | **8.82** | **97.61** | **76.79** | **66.44** | **86.27** | **60.62** |

## D.4 Analysis of Hyper-parameters Sensitivity

We show the sensitivity of hyper-parameters $\alpha$, $\beta$ and $k$ in Figure 6. The vertical axis in the figure represents the average AUROC values across common datasets. The performance comparison in the bar plot for each hyper-parameter is reported by fixing other hyper-parameters. We see that within the range of chosen hyperparameter values, our proposed method's performance does not vary significantly, which constantly outperforms baselines by a large margin.

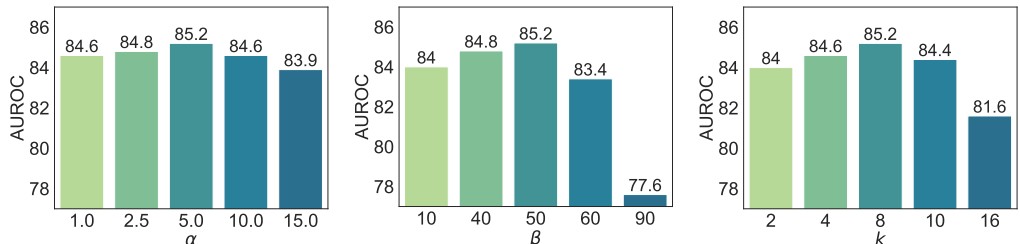

Figure 6: Sensitivity analysis of all the hyper-parameters and the averaged AUROC values of common benchmarks on model GCN with MSP are displayed. The middle bar in each plot corresponds to the hyperparameter value used in our main experiments.

## D.5 Relationship between Ratio of Intra-Edges and OOD Detection performance

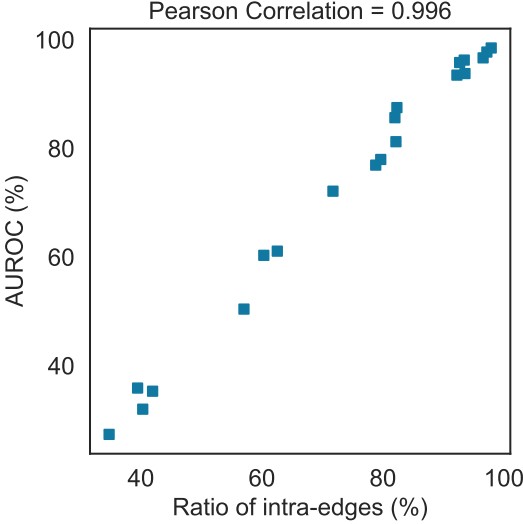

Figure 7: Ratio of intra-edges and OOD detection performance have strong positive correlation.

As shown in Figure 7, the ratio of intra-edges and OOD detection performance have strong positive correlation. This validates the correctness of Theorem 4.2. The detailed the number in Figure 7 are listed in Table 15. From the table we can see that after using our proposed augmentation, the ratio of intra-edges increases and in turn boosts the OOD detection performance.

Table 15: Ratios of intra-edges $\eta_{intra}$ and respective OOD detection performance on all benchmarks before and after employing graph augmentation.

| Datasets | Before Aug | | After Aug | |
|---|---|---|---|---|
| | $\eta_{intra}$ | AUROC | $\eta_{intra}$ | AUROC |
| cora | 82.31 | 87.52 | 92.23 | 93.50 |
| amazon | 93.44 | 96.27 | 96.56 | 96.68 |
| coauthor | 92.68 | 95.82 | 97.18 | 97.75 |
| chameleon | 57.03 | 50.42 | 78.81 | 76.93 |
| squirrel | 39.45 | 35.88 | 62.54 | 61.09 |
| reddit2 | 40.30 | 31.99 | 97.90 | 98.50 |
| ogbn-products | 81.98 | 85.66 | 93.56 | 93.79 |
| arxiv-year | 41.93 | 35.30 | 82.15 | 81.24 |
| snap-patents | 34.74 | 27.35 | 71.74 | 72.13 |
| wiki | 60.31 | 60.32 | 79.64 | 77.97 |

### D.6 Computational Complexity Analysis

Our method's computational footprint in terms of runtime and memory usage after propagation for $k$ times is:

- Time Complexity: $\mathcal{O}(N + 2k|\mathcal{E}| + n)$

- Memory Complexity: $\mathcal{O}(N + |\mathcal{E}| + n)$

While our algorithm introduces the fully connected matrix $E$, which brings about explosive growth of connections after each propagation and results in challenges in performing related computations, we adeptly transform the complex and time-consuming structural computations into simple matrix-vector multiplication operations, making original expensive operation tractable and leading to efficient linear complexity. Below, we provide the support from both mathmatical and empirical points.

**Complexity analysis.** Our method consists of two major computational modules, specifically, $(a)$ calculating connections between training nodes and testing nodes; $(b)$ augmentation propagation.

$(a)$ *Calculating connections between training nodes and testing nodes.* We utilize two indicator vectors, $I_{id}$ and $I_{ood}$, both with a length of nodes number $N$, to represent the number of connections each node has with $S_{id}$ and $S_{ood}$, respectively. The initial values of $I_{id}$ are set to 1 at the indices corresponding to $S_{id}$ and 0 at the remaining indices (corresponding to $S_{ood}$). The initial value of $I_{id}$ is denoted as $I_{id}^{(0)}$. Similarly, $I_{ood}$ has a initial value of 1 at all indices corresponding to $S_{ood}$ and 0 at the indices corresponding to $S_{id}$. We also represent the initial value of $I_{ood}$ as $I_{ood}^{(0)}$. After propagation for $k$ times, the connections between each node with $S_{id}$ are:

$$\begin{aligned}
I_{id}^{(k)} &= A^k I_{id}^{(0)} \\
&= A^{k-1} \cdot A I_{id}^{(0)} \\
&= A^{k-1} \cdot I_{id}^{(1)} \\
&= (\text{run } k - 1 \text{ times ...})
\end{aligned}$$

In the above equation, $A I_{id}^{(i)}$ can be computed in $\mathcal{O}(|\mathcal{E}_{id}|)$ with sparse matrix multiplication for each propagation, where $|\mathcal{E}_{id}|$ represents number of edges connected to $S_{id}$ nodes.

The above abalysis and conclusion also applys to computing $I_{ood}^{(k)}$, resulting computational complexity $\mathcal{O}(|\mathcal{E}_{ood}|)$, where $|\mathcal{E}_{ood}|$ is number of edges connected to $S_{ood}$ nodes. This leads to time and memory costs $\mathcal{O}(|\mathcal{E}_{id}| + |\mathcal{E}_{ood}|) = \mathcal{O}(|\mathcal{E}|)$ for calculating needed number of connections in each propagation and $\mathcal{O}(k|\mathcal{E}|)$ and $\mathcal{O}(|\mathcal{E}|)$ for time and memory costs after propagation for $k$ times, respectively.

$(b)$ *Augmentation propagation.* As described by Equation 6 in the main paper, given a raw OOD scoring vector $\hat{\mathbf{g}} \in \mathbb{R}^N$, the propagated scoring vector using augmented adjacency matrix after

propagation for $k$ times is given by:

$$\begin{aligned}
\mathbf{g}_{GRASP} &= (\bar{A}_+)^k \hat{\mathbf{g}} \\
&= (D_+^{-1} A_+)^k \hat{\mathbf{g}} \\
&= (D_+^{-1} A_+)^{k-1} \cdot (D_+^{-1}(A + E))\hat{\mathbf{g}} \\
&= (D_+^{-1} A_+)^{k-1} \cdot (D_+^{-1} \boxed{A\hat{\mathbf{g}}} + D_+^{-1} \boxed{E\hat{\mathbf{g}}}) \\
&= (\text{run } k - 1 \text{ times ...})
\end{aligned}$$

In the above equation,

- $E\hat{\mathbf{g}}$ can be computed with time/space complexity $\mathcal{O}(n)$ by a simple summation operation ($\mathbf{g}_i$ in $G$ will be replaced by $\sum_{i \in G} \mathbf{g}_i$ ) to get rid of the matrix multiplication.
- $A\hat{\mathbf{g}}$ can be computed in $\mathcal{O}(|\mathcal{E}|)$ with sparse matrix multiplication.
- $D_+^{-1}$ is an $\mathcal{O}(N)$ operation since it is scaling over all elements in the vector.

Above all, the time/memory complexity of augmented propagation is $\mathcal{O}(N + |\mathcal{E}| + n)$ for each propagation and the time and memory costs are $\mathcal{O}(N + k|\mathcal{E}| + n)$ and $\mathcal{O}(N + |\mathcal{E}| + n)$ for $k$ times respectively with memory reutilization.

Integrating the above two considerations, the time complexity of our algorithm after propagation for $k$ times is $\mathcal{O}(N + 2k|\mathcal{E}| + n)$, while the space complexity is $\mathcal{O}(N + |\mathcal{E}| + n)$. Notably, the utilization of space reuse results in a linear space complexity.

**Empirical results.** Then we conduct comprehensive experiments comparing the time and space costs of our algorithm with various baselines (both post-hoc and training-based approaches) in Table 16 and Table 17. From the experimental results, it is evident that our algorithm is highly efficient across all datasets.

Based on the experimental results in Table 16 and Table 17, we have:

- In comparison to training-based methods, post-hoc methods exhibit significantly lower runtime and memory consumption.
- Considering that the minimum time and space complexity required to run a graph algorithm is $\mathcal{O}(N + |\mathcal{E}|)$, as outlined in our algorithm complexity analysis, our algorithm incurs limited additional time and space costs, specifically $\mathcal{O}(|\mathcal{E}|)$ and $\mathcal{O}(n)$ on time and memory costs respectively in each propagation. This can be validated by the small extra overhead incurred by our algorithm compared to MSP or Energy from the table.
- Compared with training-based methods, our algorithm demonstrates substantially lower time and space demands on five large-scale datasets. And when compared to post-hoc baselines on these datasets, the overhead of our method is also reasonable. The performance of our method on these large-scale datasets underscores the strong practicality of our approach.

Table 16: Time (s) and Memory (M) costs of all algorithms with backbone GCN on common benchmarks.

| Method | Cora | | Amazon | | Coauthor | | Chameleon | | Squirrel | |
| --- | --- | --- | --- | --- | --- | --- | --- | --- | --- | --- |
| | time | memory | time | memory | time | memory | time | memory | time | memory |
| GKDE | 26.72038 | 3366.0078 | 160.00094 | 3388.5859 | 1004.73209 | 4199.8438 | 37.24027 | 3310.9219 | 170.25729 | 3234.3828 |
| GPN | 28.63986 | 3702.5703 | 58.71772 | 3719.0977 | 77.55155 | 3634.6562 | 50.64548 | 3620.9688 | 50.69732 | 3650.5781 |
| OODGAT | 77.98510 | 3369.0039 | 395.87711 | 3390.8789 | 411.43511 | 3851.3672 | 182.10900 | 3291.6289 | 454.62221 | 3319.9883 |
| MSP | 0.02508 | 622.7852 | 0.03915 | 635.4102 | 0.13018 | 1092.0469 | 0.04835 | 634.0508 | 0.05129 | 657.9961 |
| Energy | 0.04687 | 624.1016 | 0.09624 | 636.5156 | 0.11084 | 1094.7227 | 0.10995 | 634.7500 | 0.11732 | 660.5039 |
| KNN | 0.24068 | 630.9219 | 0.12629 | 658.2109 | 0.29778 | 1101.2969 | 0.17183 | 636.4375 | 0.14258 | 660.1758 |
| ODIN | 0.01500 | 621.8164 | 0.04800 | 636.4023 | 0.07353 | 1093.6406 | 0.16158 | 635.4062 | 0.06357 | 659.5781 |
| Mahalanobis | 0.14636 | 627.3945 | 0.22022 | 641.2070 | 0.28823 | 1099.3750 | 0.20611 | 637.1797 | 0.18573 | 661.4375 |
| GNNSafe | 0.05700 | 626.2344 | 0.14713 | 652.7969 | 0.34739 | 1106.1875 | 0.12293 | 637.9531 | 0.10340 | 662.4766 |
| **GRASP(ours)** | 0.05305 | 628.7930 | 0.14779 | 677.2617 | 0.25272 | 1118.0156 | 0.14554 | 644.0938 | 0.12774 | 668.0742 |

Table 17: Time (s) and Memory (M) costs of all algorithms with backbone GCN on large-scale benchmarks.

| Method | reddit2 | | ogbn-products | | arxiv-year | | snap-patents | | wiki | |
|---|---|---|---|---|---|---|---|---|---|---|
| | time | memory | time | memory | time | memory | time | memory | time | memory |
| GKDE | - | OOM | - | OOM | OOT | - | - | OOM | - | OOM |
| GPN | - | OOM | - | OOM | 444.83901 | 3808.7227 | - | OOM | - | OOM |
| OODGAT | - | OOM | - | OOM | 1214.97670 | 3491.1055 | - | OOM | - | OOM |
| MSP | 0.26172 | 2588.9844 | 0.4599 | 4571.1289 | 0.35540 | 726.5898 | 8.91248 | 4014.2188 | 431.68647 | 9765.8008 |
| Energy | 0.26987 | 2590.5273 | 0.49488 | 4567.3398 | 0.51500 | 726.1719 | 14.84026 | 4039.9102 | 296.09063 | 9788.4453 |
| KNN | 3.48512 | 2697.3789 | 4247.99327 | 4699.3047 | 3.77242 | 732.8984 | 6241.00550 | 3933.3164 | 3180.91959 | 9812.8906 |
| ODIN | 0.28509 | 2587.1289 | 0.49385 | 4562.1719 | 0.61390 | 727.6797 | 3.60580 | 4005.2305 | 244.30793 | 9760.2422 |
| Mahalanobis | 1.95486 | 2703.2734 | - | OOM | 1.27940 | 730.4141 | 5.92890 | 4056.0820 | 243.92857 | 9829.1562 |
| GNNSafe | 0.85545 | 2621.5547 | 4.19187 | 4589.9609 | 0.97220 | 790.1992 | 17.00348 | 4187.6680 | 365.39017 | 9655.1055 |
| **GRASP(ours)** | 1.86865 | 2628.3633 | 9.62959 | 4647.5273 | 0.67847 | 768.2070 | 20.97008 | 4279.9219 | 649.71668 | 9724.7852 |

## D.7 Explore How Different Propagation Mechanisms Impact the Findings.

In this section, we investigate the effect of different propagation mechanisms on graph OOD detection performance, like higher-order diffusion and the other various classical propagation mechanisms in the literature, including Personalized PageRank (PPR) [35], Heat Kernel Diffusion (GraphHeat) [80], Graph Diffusion Convolution (GDC) [20], Mixing Higher-Order Propagation (MixHop) [1], and Generalized PageRank (GPR) [10]. The emperical observations are as follows:

- From the analysis of the hyper-parameter (order of propagation) and its impact on the AUROC in Figure 6, we find that OOD detection performance benefits from propagation orders within a reasonable range. However, excessive propagation (greater than 16) may be detrimental.

- The AUROC results of the other various classical propagation mechanisms on graph OOD detection are shown in Table 18 and Table 19. We can see that these propagation mechanisms perform inconsistently among various datasets. In contrast, our propagation mechanism GRASP constantly demonstrates superior performance among all datasets.

Table 18: Various classical propagation mechanisms on each dataset.

| Method | Cora | Amazon | Coauthor | Chameleon |
|---|---|---|---|---|
| PPR | 91.98 | 95.45 | 97.29 | 49.22 |
| GraphHeat | 63.72 | 63.39 | 67.52 | 50.67 |
| GDC | 89.33 | 90.47 | 95.29 | 59.91 |
| MixHop | 91.49 | 94.42 | 97.09 | 49.33 |
| GPR | 91.87 | 94.81 | 97.23 | 48.62 |
| **GRASP** | **93.50** | **96.68** | **97.75** | **76.93** |

Table 19: Various classical propagation mechanisms on each dataset.

| Method | Squirrel | arXiv-year | snap-patents | wiki |
|---|---|---|---|---|
| PPR | 52.31 | 56.87 | 46.70 | 34.83 |
| GraphHeat | 53.92 | 38.40 | 41.00 | OOM |
| GDC | 48.17 | OOM | OOM | OOM |
| MixHop | 51.29 | 34.41 | 28.93 | 39.22 |
| GPR | 52.49 | 34.30 | 29.66 | 36.98 |
| **GRASP** | **61.09** | **81.24** | **72.13** | **77.97** |

## E   Limitations

While the proposed method of OOD score propagation shows promise in improving OOD detection in graph learning, there are limitations worth noting. The effectiveness of the method heavily relies on the connectivity and structure of the graph. In scenarios where the graph exhibits random connectivity patterns, the propagation of OOD scores with the proposed solution may no longer be effective. Addressing these limitations will be crucial for ensuring the robustness and generalizability of the proposed OOD detection approach in graph learning.

# F   Broader Impact

This paper on detecting Out-of-Distribution (OOD) nodes in graph learning holds significant potential for various fields and applications. By addressing a salient yet under-explored challenge in graph neural networks, the proposed method of OOD score propagation offers a promising avenue for enhancing the robustness and reliability of graph-based machine learning systems. The implications extend beyond the realm of graph learning, as OOD detection is a critical component in many real-world applications, including anomaly detection, fraud detection, and network security. Improving OOD detection in graph structures can lead to more accurate and reliable decision-making in domains such as social network analysis, recommendation systems, and biological network analysis. Overall, this research has the potential to drive advancements in graph-based machine learning and contribute to the development of more robust and reliable AI systems with broader societal impact.

# G   Other Related Works

**Graph OOD Generalization.** Unlike graph OOD detection, which aims to increase the gap between ID and OOD for effective OOD detection, graph OOD generalization aims to learn the invariant aspects behind ID and OOD. By enhancing these invariant points, the gap between ID and OOD data is reduced to improve the robustness of graph models on OOD data. The distinction among different proposed methods lies in identifying different invariant factors. GTRANS [30] leverages contrastive learning to learn the optimal perturbations of features and structures to enhance the GNN model's robustness against OOD. GRAPHPATCHER [31] improves GNN's robustness against low-degree scenarios during testing by corrupting a portion of nodes and reconstructing the original structure. LiSA [85] employs variational subgraph generators to generate label-invariant subgraphs for data augmentation, thereby enhancing the robustness of GNN models. [9] leverages the stability of correlation and employs a graph decorrelation method to learn stable associations, raising the graph OOD generalization capability. [78] boosts generalization by eliminating confounding bias. [41] extrapolates structure and feature spaces to generate OOD graph data, which is then used for data augmentation to improve OOD generalization. [28] strengthens graph OOD generalization by extracting invariant subgraphs and learning invariant patterns behind graphs based on these subgraphs. [8] introduces a set of minimal assumptions for feasible invariant graph learning. [5] investigates OOD generalization under different structural shifts. [22] incorporates label and environment causal independence to discover causal subgraphs for intensifying OOD generalization. [86] improves the OOD generalization of dynamic graphs by learning spatio-temporal invariant patterns from an environment learning perspective.

