# OpenReview forum: "Revisiting Score Propagation in Graph Out-of-Distribution Detection"
_NeurIPS.cc/2024/Conference — NeurIPS 2024 poster_

### Official Review · Reviewer_hXZy · 2024-07-12

**Soundness:** 3
**Presentation:** 3
**Contribution:** 2
**Rating:** 5
**Confidence:** 4

**Summary:**

In this paper, the authors attempt to tackle the task of detecting Out-of-Distribution (OOD) nodes in graph data. To this end, the authors propose an augmentation method, namely Graph-Augmented Score Propagation (GRASP). GRASP performs the task of OOD detection in graphs by increasing the ratio of intra-edges (i.e., edges only connecting in-distribution nodes or OOD nodes). Theoretical analysis is also provided to support the effectiveness of the proposed approach. Experimental results show that GRASP can outperform existing OOD detection methods in various datasets.

**Strengths:**

1. The proposed method is introduced in an informative manner.
2. The paper is easy to follow.
3. The comparative experiments are comprehensive.
4. The hyperparameter sensitivity analysis is thorough, providing readers with valuable insights.

**Weaknesses:**

1. At the beginning of the article, the author might overly emphasize the contribution of this article, to some extent giving the impression to readers that it is the first study on graph OOD detection. However, there have already been some papers in this area.​ For instance, 1) "Learning on Graphs with Out-of-Distribution Nodes," in KDD 2022 (cited); 2) "Generalizing graph neural networks on out-of-distribution graphs," in TPAMI 2023 (not cited); 3) "Good-d: On unsupervised graph out-of-distribution detection (cited)," in WSDM 2023. While their core idea differs from this paper, their contributions should not be overlooked. Given these observations, it is highly suggested that the authors conduct broader investigations of existing works, discuss the differences between the proposed methods and other related methods, and adjust the contributions of the paper.
2. The baselines selected in the experiments may not be sufficiently novel. This makes the experimental results are not convincing enough.
3. The proposed method needs more motivation from both theoretical and practical aspects. For example, are there any observations or previous studies supporting the assumptions that edges follow a Bernoulli distribution? Are there concrete (or real-world) examples that can demonstrate the importance of the number of intra-edges and inter-edges in OOD detection?

**Questions:**

1. The authors emphasize multiple times the importance of the number of intra-edges and inter-edges in OOD detection, providing theoretical proof. However, could a concrete example be provided to illustrate this concept more intuitively?
2. Does the assumption that edges follow a Bernoulli distribution receive support in most real-world datasets, practical applications, or previous studies?
3. In Table 3, the FPR on the dataset reddit2 is significantly lower than that of its comparison models. Could the author provide further explanation for this remarkable improvement?
4. Is there any underlying pattern in the ratio of intra-edges to inter-edges when achieving the best OOD detection performance across different datasets?
5. Can authors conduct broader investigations of recent related works, explicitly discuss the differences between the proposed and other related methods, and adjust the contributions of the paper (if possible)?

**Limitations:**

The authors have discussed the limitations of the proposed method in the Appendix.

---

> ### Author Rebuttal · Authors · 2024-08-06
>
> We appreciate the reviewer's constructive feedback and insightful questions. Below, we address each point in detail:
>
> >**W1&Q5. Suggestion on providing more discussion w.r.t the exisiting graph OOD works and comments on the paper's contribution position.**
>
> We appreciate your comments and the opportunity to clarify and further discuss the positioning!
>
> Firstly, we acknowledge the contributions of previous works in this field and have made it clear from the introduction (line 29) that our approach is inspired by prior research [72]. If any part of the text inadvertently suggested that we were the first to address graph OOD detection, we are welcome with advices to eliminate possible confusion and better position our contributions.
>
> Regarding the discussion with references and existing works on graph OOD:
>
> 1. **OOD Generalization vs. OOD Detection**. The uncited paper [A] suggested by the reviewer addresses a different problem -- OOD generalization rather than OOD detection. This distinction is crucial as OOD generalization focuses on correctly classifying domain-shifted data, which involves a different kind of "OOD" than the semantic shift concerns of OOD detection. Our work specifically targets scenarios where data does not conform to any in-domain class.
> 2. **Graph-level vs. node-level**. We also note a significant body of research [B-G] that focuses on graph-level OOD detection. Our work, however, contributes to the less-explored area of node-level OOD detection, where current literature remains sparse [H-L]. Our findings help fill this gap by providing new insights and methodologies.
> 3. **Training-required vs. post hoc method**. Methodologically, our approach differs significantly from those in existing literature, which often require extensive re-training [I-L]. Our post hoc solution (demonstrated in Tables 2, 3, 15, and 16) enables effective and efficient OOD detection, facilitating easier deployment and practical application in real-world scenarios.
> 4. **New theoretical contribution**. Our paper introduces fundamental theoretical analysis, a first in the realm of node-level graph OOD detection. We elucidate key factors such as the importance of intra-edge dominance, providing a new theoretical framework that aids in understanding and addressing the unique challenges of OOD detection on graphs.
>
>
> We hope these clarifications and expansions will address the concerns raised and better articulate the value and positioning of our work in the literature.
>
> [A] Generalizing graph neural networks on out-of-distribution graphs. TPAMI'23
>
> [B] GraphDE: A Generative Framework for Debiased Learning and Out-of-Distribution Detection on Graphs. NeurIPS'22
>
> [C] Towards OOD Detection in Graph Classification from Uncertainty Estimation Perspective.ICML'22
>
> [D] On Estimating the Epistemic Uncertainty of Graph Neural Networks using Stochastic Centering. ICML'23
>
> [E] GOOD-D: On Unsupervised Graph Out-Of-Distribution Detection. WSDM'23
>
> [F] A Data-centric Framework to Endow Graph Neural Networks with Out-Of-Distribution Detection Ability. KDD'23
>
> [G] HGOE: Hybrid External and Internal Graph Outlier Exposure for Graph Out-of-Distribution Detection. MM'23
>
> [H] Energy-based out-of-distribution detection for graph neural networks. ICLR'23
>
> [I] Uncertainty aware semi-supervised learning on graph data. NeurIPS'20
>
> [J] Graph posterior network: Bayesian predictive uncertainty for node classification. NeurIPS'21
>
> [K] Learning on graphs with out-of-distribution nodes. KDD'22
>
> [L] Bounded and Uniform Energy-based Out-of-distribution Detection for Graphs. ICML'24
>
>
>
> >**W2. The baselines selected in the experiments may not be sufficiently novel. This makes the experimental results are not convincing enough.**
>
> To address this concern, we have incorporated two latest  baselines, NODESafe [A] and fDBD [B], into our experiments on both common datasets (5 datasets) and large-scale datasets (3 datasets). NODESafe aims to reduce the generation of extreme scores when traing ID models on graphs, and fDBD detects OOD samples based on their feature distances to decision boundaries. Our results reveal that the performance of these methods lags significantly behind ours due to their inability to increase the proportion of intra-edges.
>
>
>
> | Datasets | cora   |        | amazon |        | coauthor |        | chameleon |        | squirrel |        |
> |----------|--------|--------|--------|--------|----------|--------|-----------|--------|----------|--------|
> | method   | AUROC  | FPR95  | AUROC  | FPR95  | AUROC    | FPR95  | AUROC     | FPR95  | AUROC    | FPR95  |
> | NODESafe | 84.09  | 68.73  | 68.65  | 91.51  | 80.26    | 71.74  | 50.72     | 92.29  | 49.26    | 94.60  |
> | fDBD     | 56.77  | 81.56  | 73.31  | 51.87  | 63.10    | 59.68  | 50.85     | 89.85  | 53.17    | 94.78  |
> | GRASP    | 93.50  | 29.70  | 96.68  | 14.38  | 97.75    | 7.84   | 76.93     | 66.88  | 61.09    | 85.59  |
>
> | Datasets | arxiv-year |        | snap-patents |        | reddit2 |        |
> |----------|------------|--------|--------------|--------|---------|--------|
> | method   | AUROC      | FPR95  | AUROC        | FPR95  | AUROC   | FPR95  |
> | NODESafe | 47.41      | 94.92  | 51.51        | 93.02  | 43.60   | 97.11  |
> | fDBD     | 48.87      | 95.82  | 43.27        | 94.99  | 55.72   | 89.78  |
> | GRASP    | 81.24      | 73.93  | 72.13        | 75.22  | 98.50   | 2.41   |
>
> [A] Bounded and Uniform Energy-based Out-of-distribution Detection for Graphs. ICML'24
>
> [B] Fast Decision Boundary based Out-of-Distribution Detector. ICML'24

---

> ### Author Response · Authors · 2024-08-06
> **Part 2 of the rebuttal**
>
> >**W3&Q2. Justification regarding using Bernoulli distribution to model edge weights.**
>
> Fair Concern! As we are dealing with discrete graphs, where edges either exist or do not, the adjacency matrix values are binary, taking on either 0 or 1. This naturally aligns with the Bernoulli distribution, which is frequently employed in graph structure learning, as exemplified in several pertinent references in [A,B,C].
>
> [A] Learning Discrete Structures for Graph Neural Networks. ICML'19.
>
> [B] Variational Inference for Graph Convolutional Networks in the Absence of Graph Data and Adversarial Settings. NeurIPS'20.
>
> [C] Data Augmentation for Graph Neural Networks. AAAI'21.
>
> >**W3-2. Examples that demonstrate the importance of the number of intra-edges and inter-edges.**
>
> We present detailed evidence in Table 14, which outlines the relationship between the ratio of intra-edges and OOD detection performance across 10 real datasets. The data illustrate that a higher number of intra-edges is crucial for effective OOD detection.
>
> >**Q1. Request for an intuitive example showing the importance of intra-edges and inter-edges in OOD detection.**
>
> Figure 2 provides a clear, intuitive example demonstrating how variations in the number of intra-edges versus inter-edges impact OOD detection. The two graphs in Figure 2 have the same nodes, but due to the differing numbers of intra-edges and inter-edges, they yield completely opposite results after score propagation. Specifically, score propagation enhances OOD detection performance only when intra-edges dominate; otherwise, it  may in turn hurt the performance. This conclusion is supported by the emperical results of our used real datasets, as shown in Table 2, Table 3, and Table 14.
>
> >**Q3. Concern about lower FPR on the dataset Reddit2 compared to other models.**
>
> The notable decrease in False Positive Rate (FPR) observed with GRASP on the Reddit2 dataset can be attributed to an increased proportion of intra-edges, as detailed in Table 14. After propagating score along $A\_+$, the high scores of ID nodes are transferred more to $\mathcal{V}\_{uid}$ than to $\mathcal{V}\_{uood}$, resulting in higher scores for $\mathcal{V}\_{uid}$ than for $\mathcal{V}\_{uood}$, thus widening the gap between $\mathcal{V}\_{uid}$ and $\mathcal{V}\_{uood}$ and reducing the misidentification of OOD. Figure 1 in the attached pdf  (Please refer to the Common Responses section titled "Author Rebuttal by Authors" https://openreview.net/forum?id=jb5qN3212b&noteId=HVpGXjNQ0B) shows the score distribution of $\mathcal{V}\_{uid}$ and $\mathcal{V}_{uood}$ on Reddit2 before and after using GRASP respectively. The shaded area with diagonal lines represents FPR, visually illustrating the aforementioned reasons.
>
> >**Q4. Patterns in the ratio of intra-edges to inter-edges for better OOD detection performance.**
>
> The following table systematically presents the impact of the intra- to inter-edge ratio ($\eta\_{intra}$/$\eta\_{inter}$) on OOD detection performance across various datasets. The results indicate that the number of intra-edges plays a crucial role in OOD detection performance. When the ratio of intra-edges is increased to dominate, score propagation can achieve excellent OOD detection performance across different datasets.
>
> | | Before Applying GRASP |        | After Applying GRASP |        |
> |---------------|-----------------------|--------|----------------------|--------|
> | Datasets      | $\eta\_{intra}$/$\eta\_{inter}$ | AUROC  |$\eta\_{intra}$/$\eta\_{inter}$ | AUROC  |
> | cora          | 4.65                  | 87.52  | 11.87                | 93.50  |
> | amazon        | 14.24                 | 96.27  | 28.07                | 96.68  |
> | coauthor      | 12.66                 | 95.82  | 34.46                | 97.75  |
> | chameleon     | 1.03                  | 50.42  | 3.72                 | 76.93  |
> | squirrel      | 0.65                  | 35.88  | 1.67                 | 61.09  |
> | reddit2       | 0.68                  | 31.99  | 46.62                | 98.50  |
> | ogbn-products | 4.55                  | 85.66  | 14.53                | 93.79  |
> | arxiv-year    | 0.72                  | 35.30  | 4.60                 | 81.24  |
> | snap-patents  | 0.53                  | 27.35  | 2.56                 | 72.13  |
> | wiki          | 1.52                  | 60.32  | 3.91                 | 77.97  |

---

> > ### Comment · Reviewer_hXZy · 2024-08-11
> > **Thanks for the detailed responses**
> >
> > Dear Authors,
> >
> >         Thanks very much for your detailed responses. I will keep my positive score on this paper.

---

> > > ### Author Response · Authors · 2024-08-11
> > > **Appreciate Your Feedback**
> > >
> > > Dear Reviewer hXZy,
> > >
> > > We are delighted that our responses have effectively addressed your concerns. Your recognition of our efforts in the paper is greatly appreciated!
> > >
> > > Warm regards,
> > >
> > > The Authors of Submission 12283.

---

### Official Review · Reviewer_aWhs · 2024-07-13

**Soundness:** 3
**Presentation:** 3
**Contribution:** 3
**Rating:** 7
**Confidence:** 3

**Summary:**

This study investigates the effectiveness of score propagation in graph raph Out-of-Distribution (OOD) detection. It explores the conditions under which score propagation can be beneficial and proposes an edge augmentation strategy (GRASP) to improve its performance. The authors provide theoretical guarantees for GRASP and demonstrate its performance compared to existing OOD detection methods on several graph datasets.

**Strengths:**

1) The study delves into the mechanisms of score propagation and derives conditions for its effectiveness. This theoretical foundation is solid and extends the understanding of graph OOD detection.

2) The proposed GRASP method is a practical and efficient solution for improving OOD detection performance. The results showed that GRASP has achieves the SOTA performance.

3.) The paper is generally in well-written and easy to follow.

**Weaknesses:**

1) it is strongly suggested to release the source code of experiments.

2) Considering the imporvement is margin in Table 2 and 3, it is also suggested that the authors should provide a standard error with statistical significance analysis in the results.

**Questions:**

1. How does the performance of GRASP vary with different OOD distributions (e.g., uniform, normal, outliers)?

2. Can the authors provide further insights into how GRASP works and how it interprets the graph structure?

3. How does the performance of GRASP scale with the size of the graph and the number of nodes? Is there any systemic analysis?

**Limitations:**

I did not find significant limitation of this study, one point may concern me that the effectiveness of GRASP relies on the connectivity and structure of the graph. In scenarios with random connectivity patterns, the method may not be as effective..

---

> ### Author Rebuttal · Authors · 2024-08-06
>
> We appreciate the reviewer's constructive feedback and insightful questions. Below, we address each point in detail:
>
> >**W1. it is strongly suggested to release the source code of experiments.**
>
> Sure! We have made the source code available at the following link: https://anonymous.4open.science/r/GRASP-EEA3/README.md.
>
>
> >**W2. Considering the imporvement is margin in Table 2 and 3, it is also suggested that the authors should provide a standard error with statistical significance analysis in the results.**
>
>
> We appreciate your observations regarding the improvements in Table 2 and 3! We'd like to emphasize that the overall improvement of our method is substantial. Nevertheless, your suggestion about including the standard error (STD) to provide statistical significance analysis is well taken. We have updated the manuscript to include standard errors for all datasets. It is important to note that the FPR95 metric displays a relatively high standard error across methods on the Cora, Amazon, and Chameleon datasets due to their small sizes, which makes the outcomes sensitive to variations in data splits. Nonetheless, our method consistently remains competitive across all five runs, as detailed in the updated Table.
>
> | Datasets    | cora          |               | amazon        |               | coauthor     |               | chameleon    |               | squirrel     |               |
> |-------------|---------------|---------------|---------------|---------------|--------------|---------------|--------------|---------------|--------------|---------------|
> | method      | AUROC         | FPR95         | AUROC         | FPR95         | AUROC        | FPR95         | AUROC        | FPR95         | AUROC        | FPR95         |
> | MSP         | 84.56 ± 5.39  | 70.86 ± 15.88 | 89.34 ± 3.49  | 49.26 ± 10.51 | 94.34 ± 0.41 | 28.82 ± 1.94  | 57.96 ± 3.31 | 85.70 ± 7.09  | 48.51 ± 0.46 | 94.68 ± 1.01  |
> | Energy      | 85.47 ± 4.98  | 67.54 ± 22.98 | 90.28 ± 3.42  | 42.13 ± 9.96  | 95.67 ± 0.25 | 20.29 ± 1.49  | 59.20 ± 4.31 | 88.06 ± 7.50  | 45.07 ± 1.68 | 93.98 ± 1.42  |
> | KNN         | 70.94 ± 5.62  | 90.20 ± 4.35  | 84.71 ± 3.28  | 65.19 ± 7.26  | 90.13 ± 0.50 | 51.24 ± 1.83  | 57.90 ± 6.48 | 93.38 ± 5.48  | 54.68 ± 2.25 | 94.72 ± 2.84  |
> | ODIN        | 84.98 ± 5.59  | 68.41 ± 18.48 | 89.90 ± 3.65  | 44.06 ± 10.69 | 95.27 ± 0.33 | 22.59 ± 1.76  | 57.94 ± 3.75 | 85.31 ± 7.64  | 44.08 ± 0.35 | 94.17 ± 0.44  |
> | Mahalanobis | 85.48 ± 1.69  | 69.68 ± 14.60 | 75.58 ± 7.97  | 96.49 ± 5.96  | 84.98 ± 0.58 | 85.71 ± 1.82  | 53.19 ± 4.30 | 95.55 ± 2.36  | 54.99 ± 0.70 | 94.90 ± 0.51  |
> | GKDE        | 86.27 ± 2.69  | 63.71 ± 14.36 | 77.26 ± 5.54  | 81.29 ± 3.36  | 95.13 ± 0.29 | 25.48 ± 1.48  | 50.14 ± 5.50 | 92.93 ± 4.89  | 49.38 ± 3.58 | 96.71 ± 0.67  |
> | GPN         | 82.93 ± 11.20 | 58.45 ± 31.98 | 82.63 ± 5.87  | 72.95 ± 19.77 | 93.82 ± 2.63 | 34.11 ± 22.46 | 68.20 ± 6.70 | 82.25 ± 6.55  | 48.38 ± 4.43 | 95.58 ± 1.65  |
> | OODGAT      | 53.63 ± 5.13  | 94.59 ± 6.38  | 66.95 ± 16.02 | 71.34 ± 15.34 | 52.18 ± 8.26 | 96.53 ± 3.39  | 59.67 ± 6.37 | 94.43 ± 3.43  | 46.13 ± 3.10 | 95.27 ± 1.00  |
> | GNNSafe     | 87.52 ± 6.16  | 54.71 ± 31.41 | 96.27 ± 0.31  | 22.39 ± 4.90  | 95.82 ± 0.28 | 16.64 ± 1.90  | 50.42 ± 0.65 | 100.00 ± 0.00 | 35.88 ± 0.24 | 100.00 ± 0.00 |
> | GRASP       | 93.50 ± 1.65  | 29.70 ± 12.25 | 96.68 ± 0.28  | 14.38 ± 6.63  | 97.75 ± 0.18 | 7.84 ± 0.58   | 76.93 ± 4.18 | 66.88 ± 6.48  | 61.09 ± 1.49 | 85.59 ± 3.61  |
>
> | Datasets    | reddit2      |              | odbn-product |              | arxiv-year    |               | snap-patents |              | wiki         |              |
> |-------------|--------------|--------------|--------------|--------------|---------------|---------------|--------------|--------------|--------------|--------------|
> | Method      | AUROC        | FPR95        | AUROC        | FPR95        | AUROC         | FPR95         | AUROC        | FPR95        | AUROC        | FPR95        |
> | MSP         | 46.61 ± 0.66 | 96.59 ± 0.14 | 70.19 ± 0.92 | 86.87 ± 0.35 | 47.24 ± 3.70  | 95.03 ± 1.46  | 46.99 ± 0.83 | 94.31 ± 0.30 | 54.70 ± 0.68 | 95.46 ± 0.32 |
> | Energy      | 44.13 ± 0.14 | 96.77 ± 0.03 | 68.13 ± 0.38 | 85.09 ± 0.45 | 51.35 ± 5.91  | 94.10 ± 2.76  | 46.03 ± 4.59 | 96.82 ± 1.07 | 29.02 ± 2.78 | 97.31 ± 1.54 |
> | KNN         | 66.74 ± 0.55 | 90.78 ± 0.75 | 73.58 ± 1.21 | 84.22 ± 2.00 | 57.96 ± 2.19  | 95.35 ± 0.92  | 53.45 ± 0.93 | 90.54 ± 1.09 | 43.69 ± 4.83 | 93.43 ± 2.57 |
> | ODIN        | 44.69 ± 0.24 | 96.74 ± 0.07 | 68.95 ± 0.52 | 85.65 ± 0.31 | 47.36 ± 3.46  | 95.06 ± 1.47  | 45.20 ± 0.87 | 94.27 ± 0.30 | 29.91 ± 0.47 | 97.88 ± 0.18 |
> | Mahalanobis | 74.89 ± 1.01 | 71.73 ± 1.55 | OOM          |              | 59.57 ± 1.27  | 88.60 ± 1.27  | 58.50 ± 0.81 | 96.03 ± 0.22 | 67.95 ± 1.56 | 72.33 ± 2.15 |
> | GKDE        | OOT          |              | OOM          |              | OOM           |               | OOM          |              | OOM          |              |
> | GPN         | OOM          |              | OOM          |              | 50.97 ± 14.98 | 95.62 ± 3.29  | OOM          |              | OOM          |              |
> | OODGAT      | OOM          |              | OOM          |              | 59.38 ± 3.44  | 92.90 ± 0.94  | OOM          |              | OOM          |              |
> | GNNSafe     | 31.99 ± 0.26 | 99.49 ± 0.07 | 85.66 ± 1.16 | 77.86 ± 1.09 | 35.30 ± 0.06  | 100.00 ± 0.00 | 27.35 ± 0.18 | 99.92 ± 0.18 | 60.32 ± 4.51 | 72.63 ± 2.05 |
> | GRASP       | 98.50 ± 0.02 | 2.41 ± 0.09  | 93.79 ± 0.24 | 39.77 ± 1.25 | 81.24 ± 0.39  | 73.93 ± 0.60  | 72.13 ± 0.06 | 75.22 ± 0.09 | 77.97 ± 1.38 | 58.49 ± 1.07 |

---

> ### Author Response · Authors · 2024-08-06
> **Part 2 of the rebuttal**
>
> >**Q1. How does the performance of GRASP vary with different OOD distributions (e.g., uniform, normal, outliers)?**
>
> We thank reviewer for the suggestion! We manually generate 2D sample points as graph nodes, with ID samples drawn from a standard Gaussian distribution and OOD samples drawn from three different distributions: a 2D uniform distribution, a different normal distribution from ID, and an outlier distribution. The outlier samples are randomly selected points outside the region of the ID sample points. We calculate the RBF kernel similarity between pairs of sample points and constructe edges between nodes with high similarity to simulate the similarity between nodes' embeddings after training GNN.
>
> The experimental results are shown in the table below. The results indicate that GRASP performs best when the OOD is normal, worst when the OOD is uniform, and intermediate when the OOD is outlier.
>
> | OOD           | AUROC  | FPR |
> |---------------|--------|-----|
> | Before GRASP  | 56.20  | 92  |
> | uniform+GRASP | 74.08  | 62  |
> | normal+GRASP  | 97.98  | 0   |
> | outlier+GRASP | 85.81  | 47  |
>
> We are happy to provide more experiment results for reviewer's interest!
>
>
> >**Q2. Can the authors provide further insights into how GRASP works and how it interprets the graph structure?**
>
>
> Thank you for your inquiry about the rationale behind GRASP! Here's a succinct overview to aid your understanding:
>
> - **Core Principle**: The effectiveness of GRASP in enhancing graph OOD detection hinges on increasing the proportion of intra-edges before performing score propagation.
> - **Implementation**: Based on this principle, GRASP identifies a subset $G$ within the graph, characterized by having more connections to ID data than to OOD data. Additional edges within $G$ are then added to enhance the score propagration for OOD detection.
>
> Regarding your question on how GRASP interprets the graph structure:
>
> - GRASP is designed to amplify the score difference between $\mathcal{V}\_{uid}$  (nodes to be identified as ID) and $\mathcal{V}\_{uood}$ (nodes to be identified as OOD). By reinforcing connections within the subset $G$ that has stronger ties to ID data, and then conducting score propagation, it exert a greater influence on $\mathcal{V}\_{uid}$ compared to $\mathcal{V}\_{uood}$. This structured modification and targeted propagation are pivotal for enhancing the discriminative capability of the network against OOD nodes.
>
> If there is any aspect of the question that we have misunderstood, or if further clarification is required, please do not hesitate to let us know!
>
>
> >**Q3. How does the performance of GRASP scale with the size of the graph and the number of nodes? Is there any systemic analysis?**
>
>
> The 10 datasets used in our experiments cover various scales from small scale to large scale. As shown in Tables 1-3, GRASP's performance is not significantly affected by the scale or size of the datasets. Instead, its performance is more influenced by the proportion of inter-edges in the original datasets.
>
> | Datasets      | #Nodes  | #Edges    | Ratio of intra-edges | Ratio of inter-edges | AUROC  |
> |---------------|---------|-----------|----------------------|----------------------|--------|
> | cora          | 2708    | 5429      | 92.23                | 7.77                 | 93.50  |
> | amazon        | 7650    | 238162    | 96.56                | 3.44                 | 96.68  |
> | coauthor      | 18333   | 163788    | 97.18                | 2.82                 | 97.75  |
> | chameleon     | 2277    | 31421     | 78.81                | 21.19                | 76.93  |
> | squirrel      | 5201    | 198493    | 62.54                | 37.46                | 61.09  |
> | reddit2       | 232965  | 23213838  | 97.90                | 2.10                 | 98.50  |
> | ogbn-products | 2449029 | 61859140  | 93.56                | 6.44                 | 93.79  |
> | arxiv-year    | 169343  | 1166243   | 82.15                | 17.85                | 81.24  |
> | snap-patents  | 2923922 | 13975788  | 71.94                | 28.06                | 72.13  |
> | wiki          | 1925342 | 303434860 | 79.64                | 20.36                | 77.97  |

---

> ### Author Response · Authors · 2024-08-07
> **Part 3 of the rebuttal**
>
> >**Limitation. GRASP may not be as effective in scenarios with random connectivity patterns**
>
> We agree with the reviewer's point that when a graph has random connectivity, all methods will perform poorly because such a graph lacks any meaningful edge information, making it indeed impossible to work effectively. To validate this, we conduct a simple experiment in Cora. Specifically, we randomly shuffle the original edges of the graph while retaining the original features and labels for training the ID model. We then evaluate the performance of each baseline on the trained model. The results, presented in the table below, confirm that all methods achieve AUROC scores around 50%, comparable to random guessing. However, in real-world graphs, random connectivity is unlikely, ex. social network. In reality, there are connections between IDs, and they naturally tend to be linked together. Therefore, it is less practical to consider the network with random connectivity.
>
> | Method      | AUROC | FPR95 |
> |-------------|-------|-------|
> | MSP         | 53.75 | 94.00 |
> | Energy      | 53.91 | 93.91 |
> | KNN         | 50.29 | 94.56 |
> | ODIN        | 54.19 | 93.73 |
> | Mahalanobis | 54.27 | 93.86 |
> | GNNSafe     | 51.97 | 93.27 |
> | GRASP       | 51.37 | 92.73 |

---

### Official Review · Reviewer_xmGd · 2024-07-16

**Soundness:** 3
**Presentation:** 2
**Contribution:** 3
**Rating:** 5
**Confidence:** 5

**Summary:**

The paper proposes a methodology called Graph-Augmented Score Propagation to improve OOD detection performance on graphs. The key idea of the paper is an edge augmentation strategy which selectively adds edges to a subset of training nodes, which is combined with score propagation for the OOD node detection task on the graphs. Theoretical analyses is provided which links the OOD score propagation to the intra-edge vs inter-edge ratios between ID and OOD samples. Experimental results are provided on several real world graph datasets with the measurement of OOD detection metrics for demonstrating the effectiveness of their method.

**Strengths:**

1. The paper is generally well written, well motivated and easy to follow.

2. Theoretical analysis of the setting when OOD score propagation will be effective is helpful for future work in this direction, and for developing OOD detection methods for graphs.

3. The key idea of the work for OOD detection is presented as a post hoc strategy, hence the practical applicability of the method is good.

4. Results in Table 2 and 3 are convincing.

**Weaknesses:**

1. The creation of the subset G is dependent on the selection of S_id and S_ood examples which uses the MSP as a measure of for creation of these sets. However it has been well-discussed that MSP suffers from several practical issues such as overconfidence [Hendrycks and Gimpel,  ICLR ‘17], poor generalization [Lee et. al, NeurIPS ‘18] and calibration issues [Guo et al, ICML ‘17]. Because of GRASP’s dependence on the MSP for creating the set G, the overall method can be non-robust.


2. Error bars are missing from all the results, which is important for assessing the consistency of the proposed method, since the data augmentation could be significantly affected by the randomness.

**Questions:**

How would the authors ensure robustness of their detection method when the confidence could vary? (see weakness pt 1 above) Did the authors do any experiments to evalute the robustness of their method and how would they address it?

**Limitations:**

The authors mention the limitations of their work in Appendix section E and societal impact in Appendix Section F.

---

> ### Author Rebuttal · Authors · 2024-08-06
>
> We appreciate the reviewer's constructive feedback and insightful questions. Below, we address each point in detail:
>
> >**W1&Q1. Concern about using MSP score to select S_id and S_ood.**
>
> Thank you for this insightful question! We acknowledge the concern that the model can sometimes exhibit overconfidence for certain OOD nodes. However, this does not undermine the applicability of GRASP when ID nodes' MSP scores are generally higher than those of OOD nodes. As demonstrated in Figure 4, we selectively utilize the highest MSP scores for ID and the lowest for OOD. This selection strategy results in more accurate estimations of $S\_{id}$ and $S\_{ood}$.
>
> We empirically support this approach with results presented in Tables 2, 3, and 5, across five datasets (Squirrel, Reddit2, arXiv-Year, snap-patents, and Wiki). These results show significant performance improvements with GRASP, particularly where original MSP scores were ineffective, ensuring more accurate distinctions between $S_{id}$ and $S_{ood}$ in each iteration.
>
> Additionally, our method offers high flexibility—**it can integrate any existing OOD scoring function**. While we use MSP in our primary demonstrations to explain GRASP's principles, substituting MSP with other well-known OOD scoring methods in our framework also yields competitive results, as shown in Table 7. This highlights GRASP’s adaptability and robustness.
>
> >**W2. Suggestion of including error bars.**
>
> This is an excellent suggestion! We have now included the main results with standard errors for all datasets in the revised manuscript and the table below. It is important to note that the FPR95 metric displays a relatively high standard error across methods on the Cora, Amazon, and Chameleon datasets due to their small sizes, which makes the outcomes sensitive to variations in data splits. Nonetheless, our method consistently remains competitive across all five runs, as detailed in the updated table in the second part of the response.

---

> ### Author Response · Authors · 2024-08-06
> **Part 2 of the rebuttal (Table with error bar)**
>
> | Datasets    | cora          |               | amazon        |               | coauthor     |               | chameleon    |               | squirrel     |               |
> |-------------|---------------|---------------|---------------|---------------|--------------|---------------|--------------|---------------|--------------|---------------|
> | method      | AUROC         | FPR95         | AUROC         | FPR95         | AUROC        | FPR95         | AUROC        | FPR95         | AUROC        | FPR95         |
> | MSP         | 84.56 ± 5.39  | 70.86 ± 15.88 | 89.34 ± 3.49  | 49.26 ± 10.51 | 94.34 ± 0.41 | 28.82 ± 1.94  | 57.96 ± 3.31 | 85.70 ± 7.09  | 48.51 ± 0.46 | 94.68 ± 1.01  |
> | Energy      | 85.47 ± 4.98  | 67.54 ± 22.98 | 90.28 ± 3.42  | 42.13 ± 9.96  | 95.67 ± 0.25 | 20.29 ± 1.49  | 59.20 ± 4.31 | 88.06 ± 7.50  | 45.07 ± 1.68 | 93.98 ± 1.42  |
> | KNN         | 70.94 ± 5.62  | 90.20 ± 4.35  | 84.71 ± 3.28  | 65.19 ± 7.26  | 90.13 ± 0.50 | 51.24 ± 1.83  | 57.90 ± 6.48 | 93.38 ± 5.48  | 54.68 ± 2.25 | 94.72 ± 2.84  |
> | ODIN        | 84.98 ± 5.59  | 68.41 ± 18.48 | 89.90 ± 3.65  | 44.06 ± 10.69 | 95.27 ± 0.33 | 22.59 ± 1.76  | 57.94 ± 3.75 | 85.31 ± 7.64  | 44.08 ± 0.35 | 94.17 ± 0.44  |
> | Mahalanobis | 85.48 ± 1.69  | 69.68 ± 14.60 | 75.58 ± 7.97  | 96.49 ± 5.96  | 84.98 ± 0.58 | 85.71 ± 1.82  | 53.19 ± 4.30 | 95.55 ± 2.36  | 54.99 ± 0.70 | 94.90 ± 0.51  |
> | GKDE        | 86.27 ± 2.69  | 63.71 ± 14.36 | 77.26 ± 5.54  | 81.29 ± 3.36  | 95.13 ± 0.29 | 25.48 ± 1.48  | 50.14 ± 5.50 | 92.93 ± 4.89  | 49.38 ± 3.58 | 96.71 ± 0.67  |
> | GPN         | 82.93 ± 11.20 | 58.45 ± 31.98 | 82.63 ± 5.87  | 72.95 ± 19.77 | 93.82 ± 2.63 | 34.11 ± 22.46 | 68.20 ± 6.70 | 82.25 ± 6.55  | 48.38 ± 4.43 | 95.58 ± 1.65  |
> | OODGAT      | 53.63 ± 5.13  | 94.59 ± 6.38  | 66.95 ± 16.02 | 71.34 ± 15.34 | 52.18 ± 8.26 | 96.53 ± 3.39  | 59.67 ± 6.37 | 94.43 ± 3.43  | 46.13 ± 3.10 | 95.27 ± 1.00  |
> | GNNSafe     | 87.52 ± 6.16  | 54.71 ± 31.41 | 96.27 ± 0.31  | 22.39 ± 4.90  | 95.82 ± 0.28 | 16.64 ± 1.90  | 50.42 ± 0.65 | 100.00 ± 0.00 | 35.88 ± 0.24 | 100.00 ± 0.00 |
> | GRASP       | 93.50 ± 1.65  | 29.70 ± 12.25 | 96.68 ± 0.28  | 14.38 ± 6.63  | 97.75 ± 0.18 | 7.84 ± 0.58   | 76.93 ± 4.18 | 66.88 ± 6.48  | 61.09 ± 1.49 | 85.59 ± 3.61  |
>
> | Datasets    | reddit2      |              | odbn-product |              | arxiv-year    |               | snap-patents |              | wiki         |              |
> |-------------|--------------|--------------|--------------|--------------|---------------|---------------|--------------|--------------|--------------|--------------|
> | Method      | AUROC        | FPR95        | AUROC        | FPR95        | AUROC         | FPR95         | AUROC        | FPR95        | AUROC        | FPR95        |
> | MSP         | 46.61 ± 0.66 | 96.59 ± 0.14 | 70.19 ± 0.92 | 86.87 ± 0.35 | 47.24 ± 3.70  | 95.03 ± 1.46  | 46.99 ± 0.83 | 94.31 ± 0.30 | 54.70 ± 0.68 | 95.46 ± 0.32 |
> | Energy      | 44.13 ± 0.14 | 96.77 ± 0.03 | 68.13 ± 0.38 | 85.09 ± 0.45 | 51.35 ± 5.91  | 94.10 ± 2.76  | 46.03 ± 4.59 | 96.82 ± 1.07 | 29.02 ± 2.78 | 97.31 ± 1.54 |
> | KNN         | 66.74 ± 0.55 | 90.78 ± 0.75 | 73.58 ± 1.21 | 84.22 ± 2.00 | 57.96 ± 2.19  | 95.35 ± 0.92  | 53.45 ± 0.93 | 90.54 ± 1.09 | 43.69 ± 4.83 | 93.43 ± 2.57 |
> | ODIN        | 44.69 ± 0.24 | 96.74 ± 0.07 | 68.95 ± 0.52 | 85.65 ± 0.31 | 47.36 ± 3.46  | 95.06 ± 1.47  | 45.20 ± 0.87 | 94.27 ± 0.30 | 29.91 ± 0.47 | 97.88 ± 0.18 |
> | Mahalanobis | 74.89 ± 1.01 | 71.73 ± 1.55 | OOM          |              | 59.57 ± 1.27  | 88.60 ± 1.27  | 58.50 ± 0.81 | 96.03 ± 0.22 | 67.95 ± 1.56 | 72.33 ± 2.15 |
> | GKDE        | OOT          |              | OOM          |              | OOM           |               | OOM          |              | OOM          |              |
> | GPN         | OOM          |              | OOM          |              | 50.97 ± 14.98 | 95.62 ± 3.29  | OOM          |              | OOM          |              |
> | OODGAT      | OOM          |              | OOM          |              | 59.38 ± 3.44  | 92.90 ± 0.94  | OOM          |              | OOM          |              |
> | GNNSafe     | 31.99 ± 0.26 | 99.49 ± 0.07 | 85.66 ± 1.16 | 77.86 ± 1.09 | 35.30 ± 0.06  | 100.00 ± 0.00 | 27.35 ± 0.18 | 99.92 ± 0.18 | 60.32 ± 4.51 | 72.63 ± 2.05 |
> | GRASP       | 98.50 ± 0.02 | 2.41 ± 0.09  | 93.79 ± 0.24 | 39.77 ± 1.25 | 81.24 ± 0.39  | 73.93 ± 0.60  | 72.13 ± 0.06 | 75.22 ± 0.09 | 77.97 ± 1.38 | 58.49 ± 1.07 |

---

> > ### Comment · Reviewer_xmGd · 2024-08-13
> > **Response to authors' rebuttal**
> >
> > Thanks for the rebuttal and sharing the additional results, I would encourage the authors to include them in the paper for a clearer indication of statistical significance of their results and contribution.
> >
> > Given my positive outlook on the paper, I would like to keep my score.

---

> > > ### Author Response · Authors · 2024-08-13
> > > **Appreciate Your Feedback**
> > >
> > > Dear Reviewer xmGd,
> > >
> > > Thank you for taking the time to read our rebuttal and for your positive feedback. We appreciate your suggestion and will incorporate these results in the revised version of our paper.
> > >
> > > Best regards,
> > >
> > > The Authors of Submission 12283.

---

### Official Review · Reviewer_hQev · 2024-07-30

**Soundness:** 2
**Presentation:** 3
**Contribution:** 3
**Rating:** 5
**Confidence:** 4

**Summary:**

This work aims to detect out-of-distribution (OOD) nodes on a graph by exploring useful OOD score propagation methods. It introduces a novel edge augmentation strategy, with a theoretical guarantee. The approach's superiority is empirically demonstrated, outperforming OOD detection baselines in various scenarios and settings.

**Strengths:**

S1. The paper conducts an in-depth analysis and empirically validates the beneficial conditions for  OOD score propagation in a graph.

S2. The paper introduces the GRASP methodology, which addresses the limitations of previous score propagation methods and demonstrates superior performance in node-level OOD detection tasks.

**Weaknesses:**

W1. Does the strategy of utilizing GRASP with additional edges compromise classification accuracy within the distribution? In other words, how can you balance the homogeneity of graph classification with the homogeneity of OOD detection tasks when adding edges?

W2. While this method is suitable for node-level OOD detection, it appears to be more challenging to adapt it to subgraph-level tasks or graph-level tasks.

**Questions:**

Q1. Is the classification accuracy in Appendix D.1 assessed before or after employing GRASP? Is there a comparison between the two?

Q2. Do similar cases discussed in Figure 2 exist within real-world datasets?

Q3. What is the accuracy of the edge-adding method presented in the text, specifically how many to intra-edges, and how many to inter-edges? What impact would incorrectly added edges have on the results?

**Limitations:**

Yes.

---

> ### Author Rebuttal · Authors · 2024-08-06
>
> We appreciate the reviewer's constructive feedback and insightful questions. Below, we address each point in detail:
>
> >**W1. Relationship between GRASP and the in-distribution classification accuracy.**
>
> Fair concern! GRASP is a post hoc OOD detection method that does not interfere with the classification process. Therefore, the classification accuracy of in-distribution (ID) data remains unchanged.
>
> >**W2. Extension to subgraph-level tasks or graph-level tasks.**
>
> We appreciate the reviewer's suggestion to extend our work to different tasks! While our primary focus is on the node-level setting, we are open to discussing how GRASP can be adapted for subgraph-level or graph-level tasks. One potential approach is to treat a subgraph or graph as a single node, with the similarity between subgraphs or graphs represented by the weights of edges. This adaptation would enable the application of our method to these broader contexts.
>
>
>
> >**Q1. Is the classification accuracy in Appendix D.1 assessed before or after employing GRASP? Is there a comparison between the two?**
>
> (Related to W1) The classification accuracy presented in Appendix D.1 is from the pretrained ID model before employing GRASP. Since our method is post hoc, the classification accuracy remains the same across all post hoc methods.
>
>
> >**Q2. Do similar cases discussed in Figure 2 exist within real-world datasets?**
>
> Yes, they do. Figure 2 is designed to intuitively illustrate our theoretical findings, which are authentically reflective of real-world scenarios. Our theory suggests that score propagation enhances OOD detection performance when intra-edges are predominant. This behavior is captured in real-world datasets, as supported by the empirical evidence presented in Table 14.
>
>
> >**Q3. What is the accuracy of the edge-adding method presented in the text, specifically how many to intra-edges, and how many to inter-edges? What impact would incorrectly added edges have on the results?**
>
> The accuracy of the added edges on the common datasets is shown in the table below:
>
> | Datasets  | Added Edges' ACC         |
> |-----------|-------------|
> | cora      | 0.94 |
> | amazon    | 0.94 |
> | coauthor  | 0.99 |
> | chameleon | 0.80 |
> | squirrel  | 0.80 |
>
> We investigate the impact on GRASP's performance by adding varying proportions of incorrect edges. The results are presented in the table below, with the "ratio" column indicating the proportion of incorrect edges added. The results show that GRASP's performance deteriorates significantly when the proportion of incorrect edges exceeds 0.3.
> | Datasets | cora   |        | amazon |        | coauthor |        | chameleon |        | squireel |        |
> |----------|--------|--------|--------|--------|----------|--------|-----------|--------|----------|--------|
> | ratio    | AUROC  | FPR95  | AUROC  | FPR95  | AUROC    | FPR95  | AUROC     | FPR95  | AUROC    | FPR95  |
> | 0        | 98.78  | 0.42   | 99.85  | 0.21   | 99.79    | 0.00   | 95.06     | 5.79   | 92.27    | 7.99   |
> | 0.1      | 87.04  | 54.57  | 90.86  | 47.73  | 90.03    | 39.10  | 77.91     | 55.09  | 70.04    | 66.20  |
> | 0.3      | 57.33  | 77.98  | 65.04  | 80.70  | 66.63    | 79.13  | 63.89     | 76.94  | 49.63    | 91.78  |
> | 0.5      | 42.52  | 80.94  | 37.09  | 90.60  | 40.75    | 88.16  | 45.79     | 90.72  | 23.43    | 96.29  |
> | 0.7      | 38.88  |  86.97 | 30.27  | 95.63  | 34.64    | 92.78  | 28.41     | 97.34  | 12.20    | 99.91  |

---

> > ### Comment · Reviewer_hQev · 2024-08-13
> > **Thanks for your reply**
> >
> > Thank you for your reply.
> >
> > I have concerns about your claim in the paper that "GRASP still works when raw MSP fails," based on the new experiment involving the addition of incorrect edges.
> > The results in Tables 2 and 3 indicate that MSP's performance is poor, which suggests that using MSP to select S_id and S_odd may introduce numerous errors. This could lead to incorrect edges being added to the augmented graph, which, based on the evidence that "GRASP's performance deteriorates significantly when the proportion of incorrect edges exceeds 0.3," could adversely impact the effectiveness of GRASP.
> >
> > It's unclear under what conditions your algorithm is efficient, specifically on which kinds of graph datasets and with which base models.

---

> ### Author Response · Authors · 2024-08-14
> **Response to Reviewer's Concerns - Part 1**
>
> Dear Reviewer hQev,
>
> Thank you for raising the concerns regarding our claim that "GRASP still works when raw MSP fails." We appreciate the opportunity to further clarify this aspect of our study.
>
> To delve deep into this, we conducted a visualization analysis of the MSP score distribution similar as Figure 4 across each dataset. Due to the limitations of the OpenReview platform, we are unable to upload these new figures to share the insight. (We will include in the revision) In these figures, we observe multiple distinct "heaps" in the score distribution. It is important to note that since MSP measures across the entire test set, it does not effectively capture all the local characteristics. Notably, our results indicate that selections made using Sid, which prioritizes the highest MSP scores, more composed of ID nodes rather than OOD nodes. This tendency is further reinforced with additional propagation iterations.
>
> We also wish to highlight additional evidence supporting the effectiveness of GRASP:
>
> 1. **GRASP is also compatible with other scoring function beyond MSP.** Substituting MSP with other well-known OOD scoring methods in our framework also yields competitive results, as shown in the table (AUROC) below.
>
> | method       | cora   | amazon | coauthor | chameleon | squirrel |
> |--------------|--------|--------|----------|-----------|----------|
> | MSP          | 84.56  | 89.34  | 94.34    | 57.96     | 48.51    |
> | MSP+prop     | 88.02  | 95.32  | 97.15    | 50.35     | 36.21    |
> | MSP+**GRASP**   | **93.50** | **96.68** | **97.75** | **76.93** | **61.09**    |
> | Energy       | 85.47  | 90.28  | 95.67    | 59.20     | 45.07    |
> | Energy+prop  | 87.52  | 96.27  | 95.82    | 50.42     | 36.49    |
> | Energy+**GRASP** | **88.34** | **96.35** | **96.64** | **62.04** | **60.66**    |
> | KNN          | 70.94  | 84.71  | 90.13    | 57.90     | 54.68    |
> | KNN+prop     | 73.70  | 92.36  | 95.47    | 49.76     | 53.99    |
> | KNN+**GRASP** | **91.48** | **97.43** | **96.52** | **76.32** | **60.24**    |
>
>
> 2. **Enhancement of Intra-edge Ratios During Propagation.** By utilizing Sid and Sood estimates, GRASP significantly increases the proportion of intra-edges, thereby improving the overall accuracy of edge addition. The ablation study results displayed below illustrate the increased accuracy of added edges when GRASP is employed, compared to scenarios where it is not used:
>
> | Datasets  | w/o GRASP | w GRASP |
> |-----------|-----------|---------|
> | cora      | 0.89      | 0.94    |
> | amazon    | 0.92      | 0.94    |
> | coauthor  | 0.93      | 0.99    |
> | chameleon | 0.51      | 0.80    |
> | squirrel  | 0.39      | 0.80    |

---

> ### Author Response · Authors · 2024-08-14
> **Response to Reviewer's Concerns - Part 2**
>
> Regarding the impact of other based models, we have shown in Table 13 that GRASP achieves optimal performance on various base models in the literature. For the reviewer’s convenience, we have included the relevant results below and in the third part of the response, which clearly attests to the versatility and practicality of GRASP:
>
> |  | Datasets | cora   |        | amazon |        | coauthor |        | chameleon |        | squirrel |        |
> |----------|-------------|--------|--------|--------|--------|----------|--------|-----------|--------|----------|--------|
> | Backbone |  Method     | FPR95  | AUROC  | FPR95  | AUROC  | FPR95    | AUROC  | FPR95     | AUROC  | FPR95    | AUROC  |
> | GAT      | MSP         | 55.33  | 88.82  | 29.88  | 94.39  | 28.15    | 94.26  | 91.27     | 61.94  | 95.21    | 47.50  |
> |          | Energy      | 80.71  | 79.16  | 26.48  | 95.24  | 20.96    | 95.65  | 92.71     | 61.11  | 96.47    | 45.69  |
> |          | KNN         | 71.14  | 81.28  | 46.42  | 90.74  | 42.51    | 91.51  | 89.02     | 61.13  | 95.37    | 53.16  |
> |          | ODIN        | 55.27  | 89.06  | 26.92  | 94.89  | 24.61    | 94.95  | 90.83     | 62.89  | 96.11    | 45.68  |
> |          | Mahalanobis | 67.92  | 86.37  | 14.28  | 95.80  | 26.27    | 94.46  | 95.35     | 50.65  | 91.36    | 57.67  |
> |          | GNNSafe     | 58.97  | 85.64  | 29.12  | 93.16  | 25.41    | 93.91  | 100.00    | 50.39  | 100.00   | 36.21  |
> |          | **GRASP**  | **22.76**  | **94.28**  | **14.21**  | **96.79**  | **8.59**     | **97.51**  | **70.15**     | **73.40**  | **85.84**    | **61.18**  |
> | GCNJK    | MSP         | 81.33  | 80.40  | 32.45  | 94.64  | 26.43    | 94.44  | 86.42     | 68.19  | 94.93    | 51.83  |
> |          | Energy      | 96.56  | 70.16  | 40.90  | 93.80  | 18.75    | 95.75  | 91.92     | 65.16  | 95.36    | 49.68  |
> |          | KNN         | 90.98  | 73.81  | 64.47  | 85.18  | 50.95    | 89.98  | 94.45     | 59.04  | 94.64    | 53.49  |
> |          | ODIN        | 81.04  | 80.68  | 28.35  | 95.13  | 21.12    | 95.41  | 86.03     | 68.58  | 95.04    | 50.64  |
> |          | Mahalanobis | 60.84  | 86.20  | 61.61  | 87.11  | 83.04    | 87.34  | 87.23     | 66.61  | 91.52    | 57.24  |
> |          | GNNSafe     | 65.01  | 83.11  | 22.41  | 96.28  | 13.27    | 96.47  | 100.00    | 50.40  | 100.00   | 36.21  |
> |          | **GRASP**  | **29.69**  | **92.98**  | **12.66**  | **96.86**  | **8.03** | **97.74** | **59.61** | **75.78** | **86.02** | **60.70**  |
> | GATJK    | MSP         | 69.56  | 84.51  | 47.21  | 91.32  | 24.66    | 95.37  | 94.39     | 55.43  | 94.67    | 50.98  |
> |          | Energy      | 62.27  | 85.75  | 34.75  | 92.89  | 17.23    | 96.38  | 91.11     | 59.01  | 95.61    | 48.76  |
> |          | KNN         | 82.54  | 74.32  | 70.98  | 83.48  | 38.95    | 92.56  | 92.21     | 61.14  | 95.20    | 54.32  |
> |          | ODIN        | 64.25  | 85.21  | 39.29  | 92.19  | 18.16    | 96.30  | 93.56     | 56.10  | 95.24    | 48.62  |
> |          | Mahalanobis | 79.60  | 79.33  | 52.79  | 88.53  | 34.60    | 93.68  | 91.59     | 52.38  | 91.52    | 56.19  |
> |          | GNNSafe     | 44.43  | 90.01  | 22.46  | 95.45  | 17.54    | 95.32  | 100.00    | 50.39  | 100.00   | 36.15  |
> |          | **GRASP** | **29.04** | **92.57** | **14.78** | **96.70** | **8.32** | **97.70** | **78.65** | **71.09** | **85.88** | **61.17**  |
> | APPNP    | MSP         | 59.37  | 89.01  | 64.64  | 86.51  | 18.38    | 96.45  | 94.24     | 48.87  | 94.41    | 50.91  |
> |          | Energy      | 81.82  | 81.21  | 62.87  | 84.36  | 14.57    | 97.01  | 90.55     | 55.75  | 90.91    | 53.04  |
> |          | KNN         | 75.33  | 81.21  | 49.55  | 89.76  | 38.44    | 91.71  | 92.14     | 54.19  | 94.12    | 53.14  |
> |          | ODIN        | 56.72  | 89.47  | 60.67  | 86.76  | 15.02    | 96.98  | 94.63     | 50.71  | 94.41    | 50.60  |
> |          | Mahalanobis | 73.64  | 86.02  | 98.75  | 62.13  | 30.20    | 93.91  | 92.38     | 58.15  | 93.29    | 56.65  |
> |          | GNNSafe     | 59.70  | 85.45  | 19.26  | 95.08  | 12.10    | 96.60  | 100.00    | 50.45  | 100.00   | 36.24  |
> |          | **GRASP** | **26.45** | **94.16** | **5.69** | **97.11** | **8.69** | **97.59** | **83.41** | **63.02** | **86.42** | **60.76**  |

---

> ### Author Response · Authors · 2024-08-14
> **Response to Reviewer's Concerns - Part 3**
>
> |  | Datasets | cora   |        | amazon |        | coauthor |        | chameleon |        | squirrel |        |
> |----------|-------------|--------|--------|--------|--------|----------|--------|-----------|--------|----------|--------|
> | Backbone |  Method     | FPR95  | AUROC  | FPR95  | AUROC  | FPR95    | AUROC  | FPR95     | AUROC  | FPR95    | AUROC  |
> | H2GCN    | MSP         | 67.00  | 86.50  | 59.23  | 86.88  | 99.37    | 40.35  | 91.00     | 62.79  | 94.34    | 57.21  |
> |          | Energy      | 68.06  | 86.84  | 57.05  | 86.21  | 97.85    | 51.65  | 92.66     | 63.24  | 96.75    | 53.18  |
> |          | KNN         | 80.00  | 79.68  | 63.85  | 80.54  | 60.66    | 77.25  | 95.13     | 56.89  | 95.62    | 57.45  |
> |          | ODIN        | 65.21  | 87.10  | 56.25  | 86.97  | 99.43    | 41.58  | 91.07     | 63.52  | 95.08    | 55.69  |
> |          | Mahalanobis | 81.67  | 80.55  | 86.26  | 77.33  | 97.92    | 61.02  | 97.62     | 58.29  | 96.36    | 53.54  |
> |          | GNNSafe     | 43.97  | 88.83  | 33.40  | 90.87  | 93.00    | 43.23  | 100.00    | 50.35  | 100.00   | 36.26  |
> |          | **GRASP** | **33.54** | **92.63** | **16.57** | **96.48** | **14.23** | **96.08** | **66.38** | **74.72** | **86.04** | **60.83**  |
> | MixHop   | MSP         | 83.94  | 78.60  | 53.56  | 90.97  | 48.66    | 90.91  | 92.95     | 56.77  | 95.60    | 49.07  |
> |          | Energy      | 83.67  | 77.15  | 57.04  | 89.28  | 28.49    | 94.67  | 94.10     | 57.21  | 95.61    | 48.87  |
> |          | KNN         | 93.36  | 69.93  | 65.41  | 86.45  | 62.40    | 85.91  | 89.52     | 57.64  | 93.44    | 54.00  |
> |          | ODIN        | 83.14  | 79.10  | 50.00  | 91.25  | 41.39    | 92.65  | 93.45     | 56.48  | 95.68    | 47.58  |
> |          | Mahalanobis | 82.35  | 80.04  | 90.05  | 81.85  | 47.41    | 91.67  | 93.93     | 56.30  | 91.33    | 56.56  |
> |          | GNNSafe     | 66.86  | 83.77  | 39.72  | 93.54  | 33.83    | 92.46  | 100.00    | 50.35  | 100.00   | 36.42  |
> |          | **GRASP** | **32.11** | **92.77** | **10.07** | **96.99** | **9.41** | **97.31** | **76.92** | **66.12** | **85.92** | **60.69**  |
> | GPR-GNN   | MSP         | 64.90  | 87.44  | 62.84  | 87.66  | 23.96    | 95.64  | 96.09     | 47.65  | 95.78    | 44.62  |
> |          | Energy      | 72.85  | 83.86  | 64.23  | 85.28  | 16.42    | 96.50  | 93.78     | 49.09  | 95.16    | 42.63  |
> |          | KNN         | 74.24  | 81.46  | 48.47  | 90.48  | 38.83    | 92.31  | 94.39     | 55.31  | 94.18    | 51.74  |
> |          | ODIN        | 62.58  | 88.13  | 55.49  | 88.41  | 17.24    | 96.51  | 96.16     | 47.50  | 95.51    | 42.32  |
> |          | Mahalanobis | 79.56  | 84.53  | 97.25  | 69.75  | 49.93    | 91.56  | 87.01     | 55.95  | 87.24    | 61.10  |
> |          | GNNSafe     | 51.65  | 85.91  | 13.63  | 96.46  | 14.73    | 95.96  | 100.00    | 50.32  | 100.00   | 36.25  |
> |          | **GRASP** | **26.71** | **94.02** | **5.30** | **97.14** | **8.28** | **97.70** | **76.53** | **72.43** | **85.40** | **61.33**  |
> | GCNII    | MSP         | 72.85  | 83.02  | 51.72  | 88.13  | 23.18    | 95.21  | 96.03     | 55.46  | 94.13    | 49.46  |
> |          | Energy      | 83.15  | 75.24  | 48.28  | 88.78  | 17.72    | 96.03  | 95.87     | 56.75  | 94.61    | 48.63  |
> |          | KNN         | 83.99  | 76.02  | 59.25  | 86.74  | 36.05    | 93.43  | 94.72     | 52.86  | 94.65    | 53.47  |
> |          | ODIN        | 71.49  | 83.31  | 49.44  | 88.35  | 19.44    | 95.75  | 95.61     | 56.63  | 94.73    | 48.34  |
> |          | Mahalanobis | 73.90  | 82.01  | 77.63  | 80.87  | 44.01    | 92.63  | 96.68     | 46.57  | 91.66    | 53.62  |
> |          | GNNSafe     | 66.70  | 83.12  | 27.08  | 93.13  | 17.87    | 94.47  | 100.00    | 50.35  | 100.00   | 36.32  |
> |          | **GRASP** | **27.92** | **93.51** | **23.53** | **93.72** | **8.82** | **97.61** | **76.79** | **66.44** | **86.27** | **60.62**  |

---

### Author Rebuttal · Authors · 2024-08-06

Dear Reviewers and ACs,

We are grateful for the insightful comments and valuable suggestions from all reviewers. In the following, we would like to summarize the contributions and revisions of this paper. As abbreviations, we refer to Reviewer hQev as R1, Reviewer xmGd as R2, Reviewer aWhs as R3, and Reviewer hXZy as R4 respectively.

**Contributions**:

- We study score propagation in-depth, theoretically elucidating the conditions under which score propagation is effective. Multiple reviewers value the theoretical contribution of our paper: ```conducts an in-depth analysis``` (R1), ```theoretical analysis is helpful for future work in this direction, and for developing OOD detection methods for graphs``` (R2), ``` theoretical foundation is solid and extends the understanding of graph OOD detection``` (R3).
- We propose a graph augmentation method aimed at increasing the ratio of intra-edges to enhance OOD detection performance without the need for training from scratch and without requiring knowledge of true OOD nodes. Multiple reviewer recognized the effectiveness  and practical values of our method, ```demonstrates superior performance``` (R1), ```results are convincing``` (R2), ```has achieves the SOTA performance``` (R3); ```the practical applicability is good``` (R2), ```the solution is practical and efficient``` (R3), ```experiments are comprehensive, the hyperparameter sensitivity analysis is thorough, providing readers with valuable insights``` (R4).
- Our paper is ```well motivated``` (R2), ```introduced in an informative manner``` (R4), ```well written``` (R2, R3), and ```easy to follow``` (R2, R3, R4).

**Responses and Revisions:**

- For Reviewer R1's concerns:
  - We clarify the relationship between GRASP and ID classification accuracy.
  - We discuss the extension of GRASP to subgraph-level and graph-level tasks.
  - We elucidate the applicability of Figure 2 in real-world datasets.
  - We investigate the impact of incorrectly added edges on the results.
- For Reviewer R2's concerns:
  - We explain the effectiveness and robustness of our method in using MSP.
  - We add error bars to the results.
- For Reviewer R3's concerns:
  - We release the source codes and pre-trained checkpoints for the reproducibility of our work.
  - We add error bars to the results.
  - We conduct additional experiments to analyze how GRASP's performance varies with different OOD distributions.
  - We provide further insights into the rationale behind GRASP and how it interprets the graph structure.
  - We analyze the relationship between GRASP and the graph size.
  - We explain the effect of random connectivity on GRASP.

- For Reviewer R4's concerns:
  - We provide a more detailed discussion of existing graph-OOD works and position our paper's contributions.
  - We add comparative experiments with two latest baselines.
  - We justify the use of the Bernoulli distribution to model edge weights.
  - We present a concrete and intuitive example to illustrate the motivation behind our approach.
  - We offer further explanation for the remarkable improvement of GRASP on FPR in reddit2.
  - We highlight the key factors in achieving optimal OOD detection performance.


Thanks again for your efforts in reviewing our work, and we hope our responses can address any concerns about this work.

Warm regards,

The Authors of Submission 12283.

---

### Decision · Program_Chairs · 2024-09-25

**Decision:**

Accept (poster)

**Comment:**

This paper theoretically understands the propagation mechanism that leads to performance degradation in OOD detection on graph data and presents a simple yet effective post-hoc methodology called GRASP to address this issue. This approach focuses on increasing the intra-edge ratio, based on the observation and theoretical insight that OOD detection becomes more difficult as the inter-edge ratio between ID and OOD nodes increases. By dividing unlabeled nodes into ID and OOD sets based on maximum softmax probability, the method identifies a subset with a high connection ratio to the ID set from labeled nodes and links the nodes within this subset. Extensive experiments demonstrate the effectiveness of the proposed method.
The reviewers initially suggested the following strong points and concerns.

Strong points:
- The paper conducts an in-depth analysis and empirically validates the beneficial conditions for OOD score propagation in a graph.

- The key idea of the work for OOD detection is presented as a post hoc strategy, hence the practical applicability of the method is good.

- The study delves into the mechanisms of score propagation and derives conditions for its effectiveness. This theoretical foundation is solid and extends the understanding of graph OOD detection.

Concerns:

- Several issues such as overconfidence and poor generalization can cause degradation of GRASP performance.

- Standard error with statistical significance analysis should be provided.

- The baselines selected in the experiments may not be sufficiently novel.

The authors have adequately addressed the reviewers' concerns, but one concern remains.
The performance of the latest baseline, NODESafe [1], presented in the authors' rebuttal differs from the results reported in the original NODESafe paper by approximately 9 in AUROC and 39 in FPR. Similarly, the baseline GNNSafe [2] included in the main table shows a discrepancy of about 5 in AUROC and 24 in FPR compared to its original paper.
Although the settings are slightly different (e.g., in the Cora dataset, the paper used 3 classes as ID and 4 classes as OOD, while NODESafe used 4 classes as ID and 3 classes as OOD, with the same GCN backbone), the performance gap seems notable even considering this. It is unclear why such a significant performance difference has occurred.

[1] Yang, Shenzhi, et al. "Bounded and Uniform Energy-based Out-of-distribution Detection for Graphs." Forty-first International Conference on Machine Learning.

[2] Wu, Qitian, et al. "Energy-based Out-of-Distribution Detection for Graph Neural Networks." The Eleventh International Conference on Learning Representations.